# Enhancing The Reliability of Out-of-distribution Image Detection in Neural Networks

**Shiyu Liang**
Coordinated Science Lab, Department of ECE
University of Illinois at Urbana-Champaign
sliang26@illinois.edu

**Yixuan Li**
Facebook Research
yixuanl@fb.com

**R. Srikant**
Coordinated Science Lab, Department of ECE
University of Illinois at Urbana-Champaign
rsrikant@illinois.edu

## Abstract

We consider the problem of detecting *out-of-distribution* images in neural networks. We propose *ODIN*, a simple and effective method that does not require any change to a pre-trained neural network. Our method is based on the observation that using temperature scaling and adding small perturbations to the input can separate the softmax score distributions between in- and out-of-distribution images, allowing for more effective detection. We show in a series of experiments that *ODIN* is compatible with diverse network architectures and datasets. It consistently outperforms the baseline approach (Hendrycks & Gimpel, 2017) by a large margin, establishing a new state-of-the-art performance on this task. For example, *ODIN* reduces the false positive rate from the baseline 34.7% to 4.3% on the DenseNet (applied to CIFAR-10 and Tiny-ImageNet) when the true positive rate is 95%.

## 1 Introduction

Modern neural networks are known to generalize well when the training and testing data are sampled from the same distribution (Krizhevsky et al., 2012; Simonyan & Zisserman, 2015; He et al., 2016; Cho et al., 2014; Zhang et al., 2017). However, when deploying neural networks in real-world applications, there is often very little control over the testing data distribution. Recent works have shown that neural networks tend to make high confidence predictions even for completely unrecognizable (Nguyen et al., 2015) or irrelevant inputs (Hendrycks & Gimpel, 2017; Szegedy et al., 2014; Moosavi-Dezfooli et al., 2017). It has been well documented (Amodei et al., 2016) that it is important for classifiers to be aware of uncertainty when shown new kinds of inputs, i.e., **out-of-distribution** examples. Therefore, being able to accurately detect out-of-distribution examples can be practically important for visual recognition tasks (Krizhevsky et al., 2012; Farabet et al., 2013; Ji et al., 2013).

A seemingly straightforward approach of detecting out-of-distribution images is to enlarge the training set of both in- and out-of-distribution examples. However, the number of out-of-distribution examples can be infinitely many, making the re-training approach computationally expensive and intractable. Moreover, to ensure that a neural network accurately classifies in-distribution samples into correct classes while correctly detecting out-of-distribution samples, one might need to employ exceedingly large neural network architectures, which further complicates the training process.

Hendrycks & Gimpel (2017) proposed a baseline method to detect out-of-distribution examples without further re-training networks. The method is based on an observation that a well-trained neural network tends to assign higher softmax scores to in-distribution examples than out-of-distribution examples. In this paper, we go further. We observe that after using temperature scaling in the softmax function (Hinton et al., 2015; Pereyra et al., 2017) and adding small controlled perturbations to inputs,

the softmax score gap between in - and out-of-distribution examples is further enlarged. We will show that the combination of these two techniques (temperature scaling and input perturbation) can lead to better detection performance. For example, provided with a pre-trained DenseNet (Huang et al., 2016) on CIFAR-10 dataset (positive samples), we test against images from TinyImageNet dataset (negative samples). Our method reduces the False Positive Rate (FPR), i.e., the fraction of misclassified out-of-distribution samples, from $34.7\%$ to $4.3\%$, when $95\%$ of in-distribution images are correctly classified. We summarize the main contributions of this paper as the following:

- We propose a simple and effective method, ODIN (Out-of-DIstribution detector for Neural networks), for detecting out-of-distribution examples in neural networks. Our method does not require re-training the neural network and is easily implementable on any modern neural architecture.

- We test ODIN on state-of-the-art network architectures (e.g., DenseNet (Huang et al., 2016) and Wide ResNet (Zagoruyko & Komodakis, 2016)) under a diverse set of in- and out-distribution dataset pairs. We show ODIN can significantly improve the detection performance, and consistently outperforms the state-of-the-art method (Hendrycks & Gimpel, 2017) by a large margin.

- We empirically analyze how parameter settings affect the performance, and further provide simple analysis that provides some intuition behind our method.

The outline of this paper is as follows. In Section 2, we present the necessary definitions and the problem statement. In Section 3, we introduce ODIN and present performance results in Section 4. We experimentally analyze the proposed method and provide some justification for our method in Section 5. We summarize the related works and future directions in Section 6 and conclude the paper in Section 7.

## 2 PROBLEM STATEMENT

In this paper, we consider the problem of distinguishing in- and out-of-distribution images on a pre-trained neural network. Let $P_{\boldsymbol{X}}$ and $Q_{\boldsymbol{X}}$ denote two distinct data distributions defined on the image space $\mathcal{X}$. Assume that a neural network $\boldsymbol{f}$ is trained on a dataset drawn from the distribution $P_{\boldsymbol{X}}$. Thus, we call $P_{\boldsymbol{X}}$ the **in-distribution** and $Q_{\boldsymbol{X}}$ the **out-distribution**, respectively. In testing, we draw new images from a mixture distribution $\mathbb{P}_{\boldsymbol{X} \times Z}$ defined on $\mathcal{X} \times \{0, 1\}$, where the conditional probability distributions $\mathbb{P}_{\boldsymbol{X}|Z=0} = P_{\boldsymbol{X}}$ and $\mathbb{P}_{\boldsymbol{X}|Z=1} = Q_{\boldsymbol{X}}$ denote in- and out-distribution respectively. Now we focus on the following problem: Given an image $\boldsymbol{X}$ drawn from the mixture distribution $\mathbb{P}_{\boldsymbol{X} \times Z}$, *can we distinguish whether the image is from in-distribution $P_{\boldsymbol{X}}$ or not*?

In this paper, we focus on detecting out-of-distribution images. However, it is equally important to correctly classify an image into the right class if it is an in-distribution image. But this can be easily done: once it has been detected that an image is in-distribution, we can simply use the original image and run it through the neural network to classify it. Thus, we do not change the predictions of the neural network for in-distribution images and only focus on improving the detection performance for out-of-distribution images.

## 3 ODIN: OUT-OF-DISTRIBUTION DETECTOR

In this section, we present our method, ODIN, for detecting out-of-distribution samples. The detector is built on two components: temperature scaling and input preprocessing. We describe the details of both components below.

**Temperature Scaling.** Assume that the neural network $\boldsymbol{f} = (f_1, ..., f_N)$ is trained to classify $N$ classes. For each input $\boldsymbol{x}$, the neural network assigns a label $\hat{y}(\boldsymbol{x}) = \arg\max_i S_i(\boldsymbol{x}; T)$ by computing the softmax output for each class. Specifically,

$$S_i(\boldsymbol{x}; T) = \frac{\exp\left(f_i(\boldsymbol{x})/T\right)}{\sum_{j=1}^{N} \exp\left(f_j(\boldsymbol{x})/T\right)}, \tag{1}$$

where $T \in \mathbb{R}^+$ is the temperature scaling parameter and set to 1 during the training. For a given input $\boldsymbol{x}$, we call the maximum softmax probability, i.e., $S_{\hat{y}}(\boldsymbol{x}; T) = \max_i S_i(\boldsymbol{x}; T)$ the **softmax score**. In this paper, we use notations $S_{\hat{y}}(\boldsymbol{x}; T)$ and $S(\boldsymbol{x}; T)$ interchangeably. Prior works have established the use of temperature scaling to distill the knowledge in neural networks (Hinton et al., 2015) and

calibrate the prediction confidence in classification tasks (Guo et al., 2017). As we shall see later, a good manipulation of temperature $T$ can push the softmax scores of in- and out-of-distribution images further apart from each other, making the out-of-distribution images distinguishable.

**Input Preprocessing.** Before feeding the image $\boldsymbol{x}$ into the neural network, we preprocess the input by adding small perturbations to it. The preprocessed image is given by

$$\tilde{\boldsymbol{x}} = \boldsymbol{x} - \varepsilon \text{sign}(-\nabla_{\boldsymbol{x}} \log S_{\hat{y}}(\boldsymbol{x}; T)), \tag{2}$$

where the parameter $\varepsilon$ can be interpreted as the perturbation magnitude. The method is inspired by the idea in the reference (Goodfellow et al., 2015), where small perturbations are added to decrease the softmax score for the true label and force the neural network to make a wrong prediction. Here, our goal and setting are rather different: we aim to increase the softmax score of any given input, without the need for a class label at all. As we shall see later, the perturbation can have stronger effect on the in- distribution images than that on out-of-distribution images, making them more separable. Note that the perturbations can be easily computed by back-propagating the gradient of the cross-entropy loss w.r.t the input.

**Out-of-distribution Detector.** The proposed approach works as follows. For each image $\boldsymbol{x}$, we first calculate the preprocessed image $\tilde{\boldsymbol{x}}$ according to the equation (2). Next, we feed the preprocessed image $\tilde{\boldsymbol{x}}$ into the neural network, calculate its softmax score $S(\tilde{\boldsymbol{x}}; T)$ and compare the score to the threshold $\delta$. We say that the image $\boldsymbol{x}$ is an in-distribution example if the softmax score is above the threshold and that the image $\boldsymbol{x}$ is an out-of-distribution example, otherwise. Therefore, the out-of-distribution detector is given by

$$g(\boldsymbol{x}; \delta, T, \varepsilon) = \begin{cases} 1 & \text{if } \max_i p(\tilde{\boldsymbol{x}}; T) \leq \delta, \\ 0 & \text{if } \max_i p(\tilde{\boldsymbol{x}}; T) > \delta. \end{cases}$$

The parameters $T, \varepsilon$ and $\delta$ are chosen so that the true positive rate (i.e., the fraction of in-distribution images correctly classified as in-distribution images) under some out-of-distribution image data set is $95\%$. (The choice of the out-of-distribution images to tune the parameters $T, \varepsilon$ and $\delta$ appears to be unimportant, as demonstrated in the appendix H.) Having chosen the parameters as above, we evaluate the performance of our algorithm using various metrics in the next section.

## 4 EXPERIMENTS

In this section, we demonstrate the effectiveness of ODIN on several computer vision benchmark datasets. We run all experiments with PyTorch[1] and we will release the code to reproduce all experimental results[2].

### 4.1 TRAINING SETUP

**Architectures and training configurations.** We adopt two state-of-the-art neural network architectures, including *DenseNet* (Huang et al., 2016) and *Wide ResNet* (Zagoruyko & Komodakis, 2016). For DenseNet, our model follows the same setup as in (Huang et al., 2016), with depth $L = 100$, growth rate $k = 12$ (Dense-BC) and dropout rate 0. In addition, we evaluate the method on a Wide ResNet, with depth 28, width 10 (WRN-28-10) and dropout rate 0. Furthermore, in Appendix A.1, we provide additional experimental results on another Wide ResNet with depth 40, width 4 (WRN-40-4). The hyper-parameters of neural networks are set identical to the original Wide ResNet (Zagoruyko & Komodakis, 2016) and DenseNet (Huang et al., 2016) implementations. All neural networks are trained with stochastic gradient descent with Nesterov momentum (Duchi et al., 2011; Kingma & Ba, 2014). Specifically, we train Dense-BC for 300 epochs with batch size 64 and momentum 0.9; and Wide ResNet for 200 epochs with batch size 128 and momentum 0.9. The learning rate starts at 0.1, and is dropped by a factor of 10 at $50\%$ and $75\%$ of the training progress, respectively.

**Accuracy.** Each neural network architecture is trained on CIFAR-10 (C-10) and CIFAR-100 (C-100) datasets (Krizhevsky & Hinton, 2009), respectively. CIFAR-10 and CIFAR-100 images are drawn from 10 and 100 classes, respectively. Both datasets consist of

| Architecture | C-10 | C-100 |
|---|---|---|
| **Dense-BC** | 4.81 | 22.37 |
| **WRN-28-10** | 3.71 | 19.86 |

Table 1: Test error rates on CIFAR-10 and CIFAR-100 datasets.

---

[1] http://pytorch.org
[2] https://github.com/facebookresearch/odin

50,000 training images and 10,000 test images. The test
error on CIFAR datasets are summarized in Table 1.

## 4.2 OUT-OF-DISTRIBUTION DATASETS

At test time, the test images from CIFAR-10 (CIFAR-100) datasets can be viewed as the in-distribution
(positive) examples. For out-of-distribution (negative) examples, we follow the setting in (Hendrycks
& Gimpel, 2017) and test on several different natural image datasets and synthetic noise datasets. All
the datasets considered are listed below.

(1) **TinyImageNet.** The Tiny ImageNet dataset[3] consists of a subset of ImageNet images (Deng
et al., 2009). It contains 10,000 test images from 200 different classes. We construct two datasets,
*TinyImageNet (crop)* and *TinyImageNet (resize)*, by either randomly cropping image patches of
size $32 \times 32$ or downsampling each image to size $32 \times 32$.

(2) **LSUN.** The Large-scale Scene UNderstanding dataset (LSUN) has a testing set of 10,000 images
of 10 different scenes (Yu et al., 2015). Similar to TinyImageNet, we construct two datasets,
*LSUN (crop)* and *LSUN (resize)*, by randomly cropping and downsampling the LSUN testing set,
respectively.

(3) **iSUN.** The iSUN (Xu et al., 2015) consists of a subset of SUN images. We include the entire
collection of 8925 images in iSUN and downsample each image to size 32 by 32.

(4) **Gaussian Noise.** The synthetic Gaussian noise dataset consists of 10,000 random 2D Gaussian
noise images, where each RGB value of every pixel is sampled from an i.i.d Gaussian distribution
with mean 0.5 and unit variance. We further clip each pixel value into the range $[0, 1]$.

(5) **Uniform Noise.** The synthetic uniform noise dataset consists of 10,000 images where each RGB
value of every pixel is independently and identically sampled from a uniform distribution on $[0, 1]$.

## 4.3 EVALUATION METRICS

We adopt the following four different metrics to measure the effectiveness of a neural network in
distinguishing in- and out-of-distribution images.

(1) **FPR at** $95\%$ **TPR** can be interpreted as the probability that a negative (out-of-distribution)
example is misclassified as positive (in-distribution) when the true positive rate (TPR) is as high as
$95\%$. True positive rate can be computed by TPR = TP / (TP+FN), where TP and FN denote true
positives and false negatives respectively. The false positive rate (FPR) can be computed by FPR =
FP / (FP+TN), where FP and TN denote false positives and true negatives respectively.

(2) **Detection Error**, i.e., $P_e$ measures the misclassification probability when TPR is 95%. The
definition of $P_e$ is given by $P_e = 0.5(1 - \text{TPR}) + 0.5\text{FPR}$, where we assume that both positive
and negative examples have the equal probability of appearing in the test set.

(3) **AUROC** is the Area Under the Receiver Operating Characteristic curve, which is also a threshold-
independent metric (Davis & Goadrich, 2006). The ROC curve depicts the relationship between
TPR and FPR. The AUROC can be interpreted as the probability that a positive example is assigned
a higher detection score than a negative example (Fawcett, 2006). A perfect detector corresponds
to an AUROC score of $100\%$.

(4) **AUPR** is the Area under the Precision-Recall curve, which is another threshold independent
metric (Manning et al., 1999; Saito & Rehmsmeier, 2015). The PR curve is a graph showing
the precision=TP/(TP+FP) and recall=TP/(TP+FN) against each other. The metric AUPR-In and
AUPR-Out in Table 2 denote the area under the precision-recall curve where in-distribution and
out-of-distribution images are specified as positives, respectively.

## 4.4 EXPERIMENTAL RESULTS

**Comparison with baseline.** In Figure 1, we show the ROC curves when DenseNet-BC-100 is
evaluated on CIFAR-10 (positive) images against TinyImageNet (negative) test examples. The red
curve corresponds to the ROC curve when using baseline method (Hendrycks & Gimpel, 2017),

---

[3] https://tiny-imagenet.herokuapp.com

| | Out-of-distribution dataset | FPR (95% TPR) ↓ | Detection Error ↓ | AUROC ↑ | AUPR In ↑ | AUPR Out ↑ |
|---|---|---|---|---|---|---|
| | | Baseline (Hendrycks & Gimpel, 2017) / ODIN | | | | |
| **Dense-BC** CIFAR-10 | TinyImageNet (crop) | 34.7/4.3 | 19.9/4.7 | 95.3/99.1 | 96.4/99.1 | 93.8/99.1 |
| | TinyImageNet (resize) | 40.8/7.5 | 22.9/6.3 | 94.1/98.5 | 95.1/98.6 | 92.4/98.5 |
| | LSUN (crop) | 39.3/8.7 | 22.2/6.9 | 94.8/98.2 | 96.0/98.5 | 93.1/97.8 |
| | LSUN (resize) | 33.6/3.8 | 19.3/4.4 | 95.4/99.2 | 96.4/99.3 | 94.0/99.2 |
| | iSUN | 37.2/6.3 | 21.1/5.7 | 94.8/98.8 | 95.9/98.9 | 93.1/98.8 |
| | Uniform | 23.5/0.0 | 14.3/2.5 | 96.5/99.9 | 97.8/100.0 | 93.0/99.9 |
| | Gaussian | 12.3/0.0 | 8.7/2.5 | 97.5/100.0 | 98.3/100.0 | 95.9/100.0 |
| **Dense-BC** CIFAR-100 | TinyImageNet (crop) | 67.8/17.3 | 36.4/11.2 | 83.0/97.1 | 85.3/97.4 | 80.8/96.8 |
| | TinyImageNet (resize) | 82.2/44.3 | 43.6/24.6 | 70.4/90.7 | 71.4/91.4 | 68.6/90.1 |
| | LSUN (crop) | 69.4/17.6 | 37.2/11.3 | 83.7/96.8 | 86.2/97.1 | 80.9/96.5 |
| | LSUN (resize) | 83.3/44.0 | 44.1/24.5 | 70.6/91.5 | 72.5/92.4 | 68.0/90.6 |
| | iSUN | 84.8/49.5 | 44.7/27.2 | 69.9/90.1 | 71.9/91.1 | 67.0/88.9 |
| | Uniform | 88.3/0.5 | 46.6/2.8 | 83.2/99.5 | 88.1/99.6 | 73.1/99.0 |
| | Gaussian | 95.4/0.2 | 50.2/2.6 | 81.8/99.6 | 87.6/99.7 | 70.1/99.1 |
| **WRN-28-10** CIFAR-10 | TinyImageNet (crop) | 38.9/23.4 | 21.9/14.2 | 92.9/94.2 | 92.5/92.8 | 91.9/94.7 |
| | TinyImageNet (resize) | 45.6/25.5 | 25.3/15.2 | 91.0/92.1 | 89.7/89.0 | 89.9/93.6 |
| | LSUN (crop) | 35.0/21.8 | 20.0/13.4 | 94.5/95.9 | 95.1/95.8 | 93.1/95.5 |
| | LSUN (resize) | 35.0/17.6 | 20.0/11.3 | 93.9/95.4 | 93.8/93.8 | 92.8/96.1 |
| | iSUN | 40.6/21.3 | 22.8/13.2 | 92.5/93.7 | 91.7/91.2 | 91.5/94.9 |
| | Uniform | 1.6/0.0 | 3.3/2.5 | 99.2/100.0 | 99.3/100.0 | 98.9/100.0 |
| | Gaussian | 0.3/0.0 | 2.6/2.5 | 99.5/100.0 | 99.6/100.0 | 99.3/100.0 |
| **WRN-28-10** CIFAR-100 | TinyImageNet (crop) | 66.6/43.9 | 35.8/24.4 | 82.0/90.8 | 83.3/91.4 | 80.2/90.0 |
| | TinyImageNet (resize) | 79.2/55.9 | 42.1/30.4 | 72.2/84.0 | 70.4/82.8 | 70.8/84.4 |
| | LSUN (crop) | 74.0/39.6 | 39.5/22.3 | 80.3/92.0 | 83.4/92.4 | 77.0/91.6 |
| | LSUN (resize) | 82.2/56.5 | 43.6/30.8 | 73.9/86.0 | 75.7/86.2 | 70.1/84.9 |
| | iSUN | 82.7/57.3 | 43.9/31.1 | 72.8/85.6 | 74.2/85.9 | 69.2/84.8 |
| | Uniform | 98.2/0.1 | 51.6/2.5 | 84.1/99.1 | 89.9/99.4 | 71.0/97.5 |
| | Gaussian | 99.2/1.0 | 52.1/3.0 | 84.3/98.5 | 90.2/99.1 | 70.9/95.9 |

Table 2: Distinguishing in- and out-of-distribution test set data for image classification. All values are percentages. ↑ indicates larger value is better, and ↓ indicates lower value is better. All parameter settings are shown in Appendix A.2. Additional results on WRN-40-4 and MNIST dataset are reported in Appendix A.1.

whereas the blue curve corresponds to our method with temperature $T = 1000$ and perturbation magnitude $\varepsilon = 0.0012$. We observe a strikingly large gap between the blue and red ROC curves. For example, when TPR= 95%, the FPR can be reduced from 34% to 4.2% by using our approach.

**Choosing parameters.** For each out-of-distribution dataset, we randomly hold out 1,000 images for tuning the parameters $T$ and $\varepsilon$. For temperature $T$, we select among 1, 2, 5, 10, 20, 50, 100, 200, 500, 1000; and for perturbation magnitude $\varepsilon$ we choose from 21 evenly spaced numbers starting from 0 and ending at 0.004. The optimal parameters are chosen to minimize the FPR at TPR 95% on the holdout set. We evaluate the our approach on the remaining test images. All parameter settings are reported in the Appendix A. We provide additional details on the effect of parameters in Section 5.

**Main results.** The main results are summarized in Table 2. For each in- and out-of-distribution dataset pair, we report both the performance of the baseline (Hendrycks & Gimpel, 2017) and our approach using temperature scaling and input preprocessing. In Table 2, we observe improved performance across all neural architectures and all dataset pairs. Noticeably, our method consistently outperforms the baseline by a large margin when measured by FPR at 95% TPR and detection error.

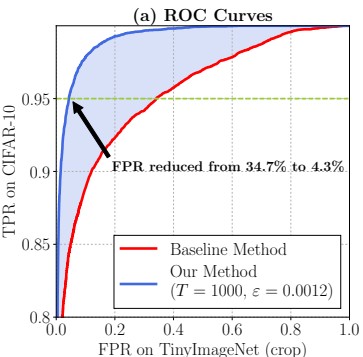

Figure 1: (a) ROC curves of baseline (red) and our method (blue) on DenseNet-BC-100 network, where CIFAR-10 and TinyImageNet (crop) are in- and out-of-distribution dataset, respectively.

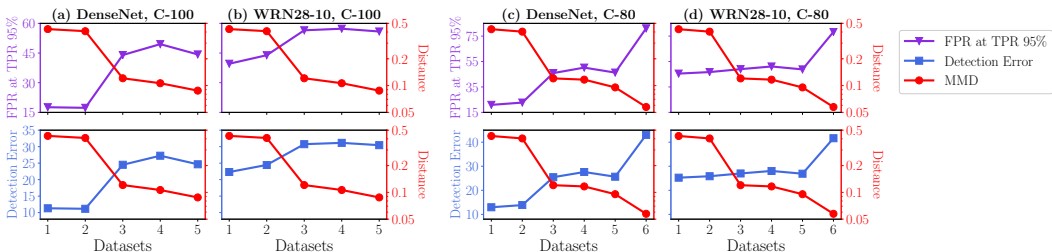

Figure 2: (a)-(d) Performance of our method vs. MMD between in- and out-of-distribution datasets. Neural networks are trained on CIFAR-100 and CIFAR-80, respectively. The out-of-distribution datasets are 1: LSUN (cop), 2: TinyImageNet (crop), 3: LSUN (resize), 4: is iSUN (resize), 5: TinyImageNet (resize) and 6: CIFAR-20.

## 4.5 EXTENSIONS

In this subsection, we analyze how the statistical distance be-
tween in- and out-of-distribution natural image dataset affects the detection performance of the proposed method.

**Data distribution distance vs. Detection performance.** To measure the statistical distance between in- and out-of-distribution datasets, we adopt a commonly used metric, maximum mean discrepancy (MMD) with Gaussian RBF kernel (Sriperumbudur et al., 2010; Gretton et al., 2012; Sutherland et al., 2016). Specifically, given two image sets, $V = \{v_1, ..., v_m\}$ and $W = \{w_1, ..., w_m\}$, the maximum mean discrepancy between $V$ and $Q$ is defined as

$$\widehat{\text{MMD}}^2(V, W) = \frac{1}{\binom{m}{2}} \sum_{i \neq j} k(v_i, v_j) + \frac{1}{\binom{m}{2}} \sum_{i \neq j} k(w_i, w_j) - \frac{2}{\binom{m}{2}} \sum_{i \neq j} k(v_i, w_j),$$

where $k(\cdot, \cdot)$ is the Gaussian RBF kernel, i.e., $k(x, x') = \exp\left(-\frac{\|x-x'\|_2^2}{2\sigma^2}\right)$. We use the same method used by Sutherland et al. (2016) to choose $\sigma$, where $2\sigma^2$ is set to the median of all Euclidean distances between all images in the aggregate set $V \cup W$.

In Figure 2 (a)(b), we show how the performance of ODIN varies against the MMD distances between in- and out-of-distribution datasets[4]. The datasets (on x-axis) are ranked in the descending order of MMD distances with CIFAR-100. There are two interesting observations can be drawn from these figures. First, we find that the MMD distances between the cropped datasets and CIFAR-100 tend to be larger. This is likely due to the fact that cropped images only contain local image context and are therefore more distinct from CIFAR-100 images, while resized images contain global patterns and are thus similar to images in CIFAR-100. Second, we observe that the MMD distance tends to be negatively correlated with the detection performance. This suggests, not surprisingly, that the detection task becomes harder as in and out-of-distribution images are more similar to each other.

**Same-manifold datasets.** Furthermore, we investigate the extreme scenario when in- and out-of-distribution datasets are on the same manifold. In experiment, we randomly split CIFAR-100 into two disjoint datasets containing 80 and 20 classes each. We name them *CIFAR-80* and *CIFAR-20*, respectively. We train both DenseNet and Wide ResNet-28-10 on the CIFAR-80 dataset (in-distribution) and evaluate the detection performance on the CIFAR-20 dataset (out-distribution). All hyperparameters used here are exactly the same as in Section 4.1. The MMD distance between CIFAR-20 and CIFAR-80 is much smaller than other dataset pairs. In Figure 2 (c)(d), we observe that both FPR at TPR 95% and detection error become larger on the CIFAR-20 dataset. This coincides with our expectation that the detection task becomes extremely hard when in- and out-of-distribution dataset locate on the same manifold. We provide additional experimental results in Appendix A.1 and Appendix G.

## 5 DISCUSSIONS

### 5.1 EFFECTS OF PARAMETERS

In this subsection, we empirically show how temperature $T$ and perturbation magnitude $\varepsilon$ affect FPR at TPR 95% and AUROC on DenseNet and Wide ResNet-28-10. Additional results on other

---

[4]All distances are provided in Appendix G.

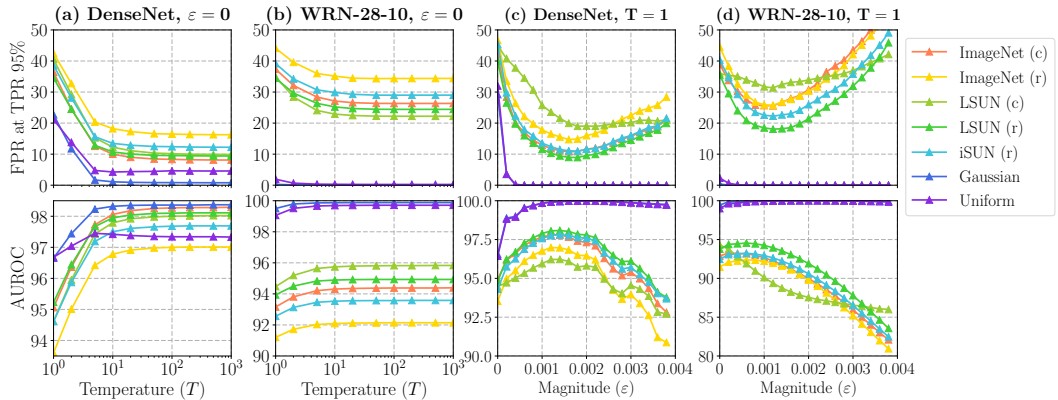

Figure 3: (a)(b) Effects of temperature $T$ when $\varepsilon = 0$. (c)(d) Effects of perturbation magnitude $\varepsilon$ when $T = 1$. All networks are trained on CIFAR-10 (in-distribution). Additional results on other metrics and Wide ResNet-40 are provided in Appendix B.

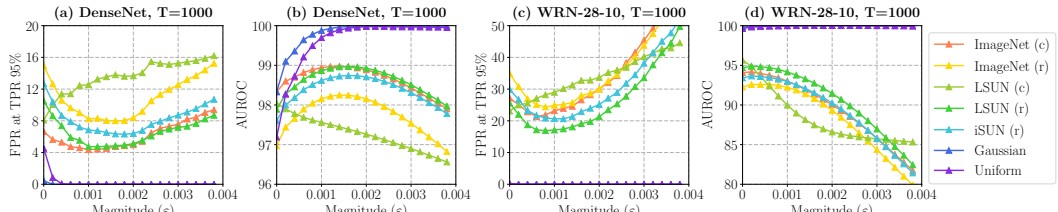

Figure 4: (a)(b) Effects of perturbation magnitude $\varepsilon$ on DenseNet when $T$ is large (e.g., $T = 1000$). (c)(d) Effects of perturbation magnitude of $\varepsilon$ on Wide-ResNet-28-10 when $T$ is large (e.g., $T = 1000$). All networks are trained on CIFAR-10. Additional results on other metrics and Wide ResNet-40 are provided in Appendix B.

metrics and architectures are provided in Appendix B. We show the detection performance when using only the temperature scaling method (see Figure 3(a)(b), $\varepsilon = 0$), or the input preprocessing method (see Figure 3(c)(d), $T = 1$). In Figure 4, we show the detection performance w.r.t $\varepsilon$ when $T$ is optimal (e.g., $T$=1000). First, from Figure 3 (a)(b), we observe that increasing the temperature can improve the detection performance, although the effects diminish when $T$ is sufficiently large (e.g., $T > 100$). Next, from Figure 3(c)(d) and Figure 4, we observe that we can further improve the detection performance by appropriately choosing the perturbation magnitudes. We can achieve overall better performance by combining both (1) temperature scaling and (2) input preprocessing.

## 5.2 Analysis on Temperature Scaling

In this subsection, we analyze the effectiveness of the temperature scaling method. As shown in Figure 3 (a) and (b), we observe that a sufficiently large temperature yields better detection performance although the effects diminish when $T$ is too large. To gain insight, we can use the Taylor expansion of the softmax score (details provided in Appendix D). When $T$ is sufficiently large, we have

$$S_{\hat{y}}(\boldsymbol{x}; T) \approx \frac{1}{N - \frac{1}{T} \sum_i [f_{\hat{y}}(\boldsymbol{x}) - f_i(\boldsymbol{x})] + \frac{1}{2T^2} \sum_i [f_{\hat{y}}(\boldsymbol{x}) - f_i(\boldsymbol{x})]^2}, \quad (3)$$

by omitting the third and higher orders. For simplicity of notation, we define

$$U_1(\boldsymbol{x}) = \frac{1}{N-1} \sum_{i \neq \hat{y}} [f_{\hat{y}}(\boldsymbol{x}) - f_i(\boldsymbol{x})] \quad \text{and} \quad U_2(\boldsymbol{x}) = \frac{1}{N-1} \sum_{i \neq \hat{y}} [f_{\hat{y}}(\boldsymbol{x}) - f_i(\boldsymbol{x})]^2. \quad (4)$$

**Interpretations of $U_1$ and $U_2$.** By definition, $U_1$ measures the extent to which the largest unnormalized output of the neural network deviates from the remaining outputs; while $U_2$ measures the extent to which the remaining smaller outputs deviate from each other. We provide formal mathematical derivations in Appendix F. In Figure 5(a), we show the distribution of $U_1$ for each out-of-distribution dataset vs. the in-distribution dataset (in red). We observe that the largest outputs of the neural network on in-distribution images deviate more from the remaining outputs. This is likely due to the fact that neural networks tend to make more confident predictions on in-distribution images.

Further, we show in Figure 5(b) the expectation of $U_2$ conditioned on $U_1$, i.e., $E[U_2|U_1]$, for each dataset. The red curve (in-distribution images) has overall higher expectation. This indicates that,

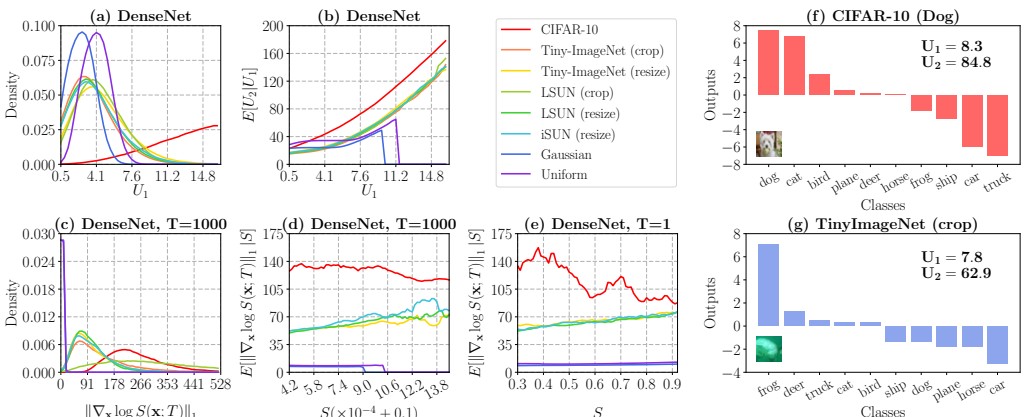

Figure 5: (a) Probability density of $U_1$ under different datasets on DenseNet. (b) Expectations of $U_2$ conditioned on $U_1$ on DenseNet. (c) Probability density of the norm of gradient on DenseNet under temperature $1,000$. (c)(d) Expectation of the norm of gradient conditioned on the softmax scores on DenseNet under temperature $T = 1000$ and $T = 1$, respectively. (f)(g) Outputs of DenseNet on each class for an image of dog from CIFAR-10 and an image from TinyImageNet (crop). The DenseNet is trained on CIFAR-10. Additional results on other architectures are provided in Appendix C.

when two images have similar values on $U_1$, the in-distribution image tends to have a much higher value of $U_2$ than the out-of-distribution image. In other words, for in-distribution images, the remaining outputs (excluding the largest output) tend to be more separated from each other compared to out-of-distribution datasets. This may happen when some classes in the in-distribution dataset share common features while others differ significantly. To illustrate this, in Figure 5 (f)(g), we show the outputs of each class using a DenseNet (trained on CIFAR-10) on a dog image from CIFAR-10, and another image from TinyImageNet (crop). For the image of dog, we can observe that the largest output for the label *dog* is close to the output for the label *cat* but is quite separated from the outputs for the label *car* and *truck*. This is likely due to the fact that, in CIFAR-10, images of dogs are very similar to the images of cats but are quite distinct from images of car and truck. For the image from TinyImageNet (crop), despite having one large output, the remaining outputs are close to each other and thus have a smaller deviation.

**The effects of** $T$**.** To see the usefulness of adopting a large $T$, we can first rewrite the softmax score function in Equation (3) as $S \propto (U_1 - U_2/2T)/T$. Hence the softmax score is largely determined by $U_1$ and $U_2/2T$. As noted earlier, $U_1$ makes in-distribution images produce larger softmax scores than out-of-distribution images since $S \propto U_1$, while $U_2$ has the exact opposite effect since $S \propto -U_2$. Therefore, by choosing a sufficiently large temperature, we can compensate the negative impacts of $U_2/2T$ on the detection performance, making the softmax scores between in- and out-of-distribution images more separable. Eventually, when $T$ is sufficiently large, the distribution of softmax score is almost dominated by the distribution of $U_1$ and thus increasing the temperature further is no longer effective. This explains why we see in Figure 3 (a)(b) that the performance does not change when $T$ is too large (e.g., $T > 100$). In Appendix E, we provide a formal proof showing that the detection error eventually converges to a constant number when $T$ goes to infinity.

## 5.3 ANALYSIS ON INPUT PREPROCESSING

As noted previously, using the temperature scaling method by itself can be effective in improving the detection performance. However, the effectiveness quickly diminishes as $T$ becomes very large. In order to make further improvement, we complement temperature scaling with input preprocessing. This has already been seen in Figure 4, where the detection performance is improved by a large margin on most datasets when $T = 1000$, provided with an appropriate perturbation magnitude $\varepsilon$ is chosen. In this subsection, we provide some intuition behind this.

To explain, we can look into the first order Taylor expansion of the log-softmax function for the perturbed image $\tilde{\boldsymbol{x}}$, which is given by

$$\log S_{\hat{y}}(\tilde{\boldsymbol{x}}; T) = \log S_{\hat{y}}(\boldsymbol{x}; T) + \varepsilon \left\| \nabla_{\boldsymbol{x}} \log S_{\hat{y}}(\boldsymbol{x}; T) \right\|_1 + o(\varepsilon),$$

where $\boldsymbol{x}$ is the original input.

**The effects of gradient.** In Figure 5 (c), we present the distribution of $\left\| \nabla_{\boldsymbol{x}} \log S(\boldsymbol{x}; T) \right\|_1$ — the 1-norm of gradient of log-softmax with respect to the input $\boldsymbol{x}$ — for all datasets. A salient observation

is that CIFAR-10 images (in-distribution) tend to have larger values on the norm of gradient than most out-of-distribution images. To further see the effects of the norm of gradient on the softmax score, we provide in Figures 5 (d) the conditional expectation $E[\|\nabla_{\boldsymbol{x}} \log S(\boldsymbol{x}; T)\|_1 | S]$. We can observe that, when an in-distribution image and an out-of-distribution image have the same softmax score, the value of $\|\nabla_{\boldsymbol{x}} \log S(\boldsymbol{x}; T)\|_1$ for in-distribution image tends to be larger.

We illustrate the effects of the norm of gradient in Figure 6. Suppose that an in-distribution image $\boldsymbol{x}_1$ (blue) and an out-of-distribution image $\boldsymbol{x}_2$ (red) have similar softmax scores, i.e., $S(\boldsymbol{x}_1) \approx S(\boldsymbol{x}_2)$. After input processing, the in-distribution image can have a much larger softmax score than the out-of-distribution image $\boldsymbol{x}_2$ since $\boldsymbol{x}_1$ results in a much larger value on the norm of softmax gradient than that of $\boldsymbol{x}_2$. Therefore, in- and out-of-distribution images are more separable from each other after input preprocessing[5].

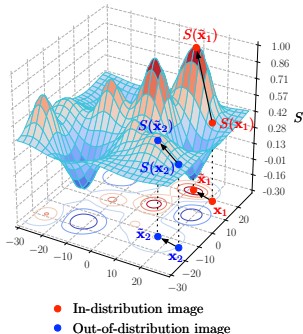

Figure 6: Illustration of effects of the input preprocessing.

**The effect of $\varepsilon$.** When the magnitude $\varepsilon$ is sufficiently small, adding perturbations does not change the predictions of the neural network, i.e., $\hat{y}(\tilde{\boldsymbol{x}}) = \hat{y}(\boldsymbol{x})$. However, when $\varepsilon$ is not negligible, the gap of softmax scores between in- and out-of-distribution images can be affected by $\|\nabla_{\boldsymbol{x}} \log S(\boldsymbol{x}; T)\|_1$. Our observation is consistent with that in (Szegedy et al., 2014; Goodfellow et al., 2015; Moosavi-Dezfooli et al., 2017), which show that the softmax scores tend to change significantly if small perturbations are added to the in-distribution images. It is also worth noting that using a very large $\varepsilon$ can lead to performance degradation, as seen in Figure 4. This is likely due to the fact that the second and higher order terms in the Taylor expansion are no longer insignificant when the perturbation magnitude is too large.

## 6 RELATED WORKS AND FUTURE DIRECTIONS

The problem of detecting out-of-distribution examples in low-dimensional space has been well-studied in various contexts (see the survey by Pimentel et al. (2014)). Conventional methods such as density estimation, nearest neighbor and clustering analysis are widely used in detecting low-dimensional out-of-distribution examples (Chow, 1970; Vincent & Bengio, 2003; Ghoting et al., 2008; Devroye et al., 2013), . The density estimation approach uses probabilistic models to estimate the in-distribution density and declares a test example to be out-of-distribution if it locates in the low-density areas. The clustering method is based on the statistical distance, and declares an example to be out-of-distribution if it locates far from its neighborhood. Despite various applications in low-dimensional spaces, unfortunately, these methods are known to be unreliable in high-dimensional space such as image space (Wasserman, 2006; Theis et al., 2015). In recent years, out-of-distribution detectors based on deep models have been proposed. Schlegl et al. (2017) train a generative adversarial networks to detect out-of-distribution examples in clinical scenario. Sabokrou et al. (2016) train a convolutional network to detect anomaly in scenes. Andrews et al. (2016) adopt transfer representation-learning for anomaly detection. All these works require enlarging or modifying the neural networks. In a more recent work, Hendrycks & Gimpel (2017) found that pre-trained neural networks can be overconfident to out-of-distribution example, limiting the effectiveness of detection. Our paper aims to improve the performance of detecting out-of-distribution examples, without requiring any change to an existing well-trained model.

Our approach leverages the following two interesting observations to help better distinguish between in- and out-of-distribution examples: (1) On in-distribution images, modern neural networks tend to produce outputs with larger variance across class labels, and (2) neural networks have larger norm of gradient of log-softmax scores when applied on in-distribution images. We believe that having a better understanding of these phenomenon can lead to further insights into this problem.

## 7 CONCLUSIONS

In this paper, we propose a simple and effective method to detect out-of-distribution data samples in neural networks. Our method does not require retraining the neural network and significantly

---

[5]Similar observation can be seen when $T = 1$, where we present the conditional expectation of the norm of softmax gradient in Figure 5 (e).

improves on the baseline (state-of-the-art) on different neural architectures across various in and out-distribution dataset pairs. We empirically analyze the method under different parameter settings, and provide some insights behind the approach. Future work involves exploring our method in other applications such as speech recognition and natural language processing.

## ACKNOWLEDGMENTS

The research reported here was supported by NSF Grant CPS ECCS 1739189.

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

# A  SUPPLEMENTARY RESULTS IN SECTION 4.4

## A.1  EXPERIMENTAL RESULTS

| | Out-of-distribution dataset | FPR (95% TPR) ↓ | Detection Error ↓ | AUROC ↑ | AUPR In ↑ | AUPR Out ↑ |
|---|---|---|---|---|---|---|
| | | Baseline (Hendrycks & Gimpel, 2017) / Ours | | | | |
| **WRN-40-4** CIFAR-10 | TinyImageNet (crop) | 49.8/36.7 | 27.4/20.9 | 87.3/89.3 | 85.1/86.7 | 87.2/90.7 |
| | TinyImageNet (resize) | 62.3/49.1 | 33.6/27.1 | 79.3/81.6 | 73.5/76.9 | 80.6/84.8 |
| | LSUN (crop) | 34.6/23.0 | 19.8/14.0 | 93.4/95.1 | 93.1/94.3 | 92.4/95.2 |
| | LSUN (resize) | 54.5/35.1 | 29.8/20.1 | 84.7/87.0 | 79.8/82.6 | 85.3/89.7 |
| | iSUN | 58.6/41.0 | 31.8/23.0 | 82.1/84.9 | 76.4/80.2 | 83.2/87.8 |
| | Uniform | 26.6/3.2 | 15.8/4.1 | 96.1/99.2 | 97.0/99.2 | 94.8/99.2 |
| | Gaussian | 21.8/0.9 | 13.4/3.0 | 96.5/99.7 | 97.5/99.7 | 94.7/99.7 |
| **WRN-40-4** CIFAR-100 | TinyImageNet (crop) | 66.9/43.3 | 36.0/24.1 | 81.3/88.5 | 80.6/87.2 | 80.1/89.1 |
| | TinyImageNet (resize) | 78.1/55.1 | 41.5/30.1 | 72.6/81.6 | 69.4/78.0 | 71.6/83.4 |
| | LSUN (crop) | 74.9/35.9 | 40.0/20.4 | 79.1/90.8 | 81.4/89.9 | 76.3/91.5 |
| | LSUN (resize) | 77.9/50.0 | 41.5/27.5 | 75.2/85.6 | 73.1/83.5 | 73.3/86.4 |
| | iSUN | 79.5/52.9 | 42.2/28.9 | 74.3/84.3 | 72.9/81.9 | 71.9/85.1 |
| | Uniform | 84.7/3.3 | 44.9/4.2 | 86.3/98.8 | 90.5/99.1 | 77.0/97.9 |
| | Gaussian | 77.2/3.1 | 41.1/4.0 | 86.4/99.0 | 90.2/99.2 | 78.6/98.6 |
| **Dense-BC** CIFAR-80 | CIFAR-20 | 84.1/81.1 | 44.9/43.0 | 76.6/77.8 | 79.4/80.6 | 71.6/73.6 |
| | TinyImageNet (crop) | 72.9/22.7 | 39.0/13.8 | 83.4/96.2 | 86.3/96.6 | 79.9/95.8 |
| | TinyImageNet (resize) | 84.4/46.3 | 44.7/25.6 | 76.8/91.7 | 80.3/92.7 | 71.5/90.4 |
| | LSUN (crop) | 67.1/20.9 | 36.0/12.9 | 84.6/96.2 | 86.9/96.4 | 82.1/96.0 |
| | LSUN (resize) | 84.9/45.9 | 45.0/25.4 | 77.5/91.8 | 81.4/92.9 | 71.6/90.2 |
| | iSUN | 86.1/50.2 | 50.5/27.6 | 76.1/90.5 | 79.8/91.3 | 69.9/88.8 |
| | Uniform | 100.0/0.9 | 52.5/3.0 | 64.3/98.6 | 78.4/99.1 | 52.2/96.6 |
| | Gaussian | 98.5/1.2 | 51.8/3.1 | 80.4/99.6 | 86.7/99.6 | 68.0/99.1 |
| **WRN-28-10** CIFAR-80 | CIFAR-20 | 80.4/78.3 | 42.7/41.6 | 79.2/80.4 | 81.5/82.2 | 74.2/76.2 |
| | TinyImageNet (crop) | 71.3/46.7 | 38.1/25.9 | 83.1/91.9 | 85.9/92.6 | 79.7/90.7 |
| | TinyImageNet (resize) | 81.0/48.8 | 43.0/26.9 | 77.1/89.2 | 80.0/89.5 | 72.6/88.5 |
| | LSUN (crop) | 74.4/45.5 | 39.7/25.2 | 82.0/92.9 | 84.4/93.0 | 78.2/91.5 |
| | LSUN (resize) | 81.9/49.0 | 43.5/27.0 | 78.8/90.1 | 82.2/90.8 | 73.4/88.8 |
| | iSUN | 82.7/51.1 | 43.9/28.1 | 78.3/89.4 | 81.5/90.0 | 72.6/88.0 |
| | Uniform | 99.6/1.4 | 52.3/3.2 | 80.6/98.9 | 87.7/99.2 | 66.8/97.6 |
| | Gaussian | 100.0/0.4 | 52.5/2.7 | 79.7/99.1 | 87.4/99.4 | 65.5/98.0 |
| **WRN-40-4** CIFAR-80 | CIFAR-20 | 82.4/78.4 | 43.7/41.7 | 76.8/78.1 | 78.9/79.1 | 72.2/75.0 |
| | TinyImageNet (crop) | 68.3/34.3 | 36.6/19.6 | 83.6/93.4 | 85.9/94.0 | 81.2/92.5 |
| | TinyImageNet (resize) | 80.6/53.5 | 42.8/29.2 | 76.2/87.7 | 78.5/88.3 | 72.5/86.1 |
| | LSUN (crop) | 72.2/33.2 | 38.6/19.1 | 83.1/93.4 | 86.3/93.7 | 79.7/93.1 |
| | LSUN (resize) | 79.1/51.2 | 42.0/28.1 | 77.6/88.8 | 80.0/89.4 | 73.9/87.3 |
| | iSUN | 81.2/53.2 | 43.1/29.1 | 76.2/87.7 | 78.7/88.3 | 72.2/86.1 |
| | Uniform | 99.7/48.6 | 52.4/26.8 | 65.6/93.8 | 77.3/95.7 | 53.7/88.7 |
| | Gaussian | 99.7/10.7 | 52.4/7.8 | 74.3/97.7 | 83.0/98.4 | 61.2/95.2 |
| **MNIST** | Omniglot | 0.2/0.0 | 2.6/2.5 | 99.6/100.0 | 99.7/100.0 | 99.5/100.0 |
| | notMNIST | 10.3/8.7 | 7.7/6.8 | 97.2/98.2 | 97.5/98.4 | 97.4/98.0 |
| | CIFAR-10bw | 0.1/0.0 | 2.5/2.5 | 99.7/100.0 | 99.8/100.0 | 99.7/100.0 |
| | Gaussian | 0.0/0.0 | 2.5/2.5 | 99.7/100.0 | 99.8/100.0 | 99.7/100.0 |
| | Uniform | 0.0/0.0 | 2.5/2.5 | 99.9/100.0 | 99.9/100.0 | 99.9/100.0 |

Table 3: Distinguishing in- and out-of-distribution test set data for image classification. All values are percentages. ↑ indicates larger value is better, and ↓ indicates lower value is better.

**MNIST:** We used the same MNIST classifier used by Hendrycks & Gimpel (2017), which is a three-layer, 256 neuron-wide, fully connected network trained for 30 epochs with Adam (Kingma & Ba, 2014). The classifier achieve **99.34%** test accuracy on the MNIST test set. We compare our method with the baseline (Hendrycks & Gimpel, 2017) on five different out-of-distribution datasets: (1) Omniglot dataset (Lake et al., 2015) contains images of handwritten characters in stead of the handwritten digits in MNIST; (2) notMNIST (Bulatov, 2011) dataset contains typeface characters;

(3) CIFAR-10bw contains black and white rescaled CIFAR-10 images; (4)(5) Gaussian and Uniform image set contains the synthetic Gaussian and Uniform noise images used in Section 4.2.

**Wide ResNet-40-4:** We use the same architecture used by Hendrycks & Gimpel (2017) to evaluate the baseline and our method. The Wide ResNet-40-4 achieves **95.7%** test accuracy on CIFAR-10 dataset and achieve **79.27%** test accuracy on CIFAR-100.

**CIFAR-80:** DenseNet-BC-100 achieves **78.94%** test accuracy on CIFAR-80, while Wide ResNet-28-10 achieves **81.71%** test accuracy and Wide ResNet-40-4 achieves **79.53%** test accuracy on CIFAR-80.

## A.2 PARAMETER SETTINGS

For MNIST, we set $T = 1000$ and $\varepsilon = 0$. The parameter settings for other structures are shown as follows.

| | DenseNet-BC-100 | | |
|---|---|---|---|
| **Out-of-distribution datasets** | **CIFAR-10** | **CIFAR-80** | **CIFAR-100** |
| **TinyImageNet (crop)** | 0.0014 | 0.002 | 0.002 |
| **TinyImageNet (resize)** | 0.0014 | 0.0022 | 0.0022 |
| **LSUN (crop)** | 0 | 0.0036 | 0.0038 |
| **LSUN (resize)** | 0.0014 | 0.002 | 0.0018 |
| **iSUN** | 0.0014 | 0.002 | 0.002 |
| **Uniform** | 0.0014 | 0.0028 | 0.0024 |
| **Gaussian** | 0.0014 | 0.0026 | 0.0028 |
| **CIFAR-20** | - | 0.0002 | - |

Table 4: Optimal perturbation magnitude $\varepsilon$ for reproducing main results in Table 2 and 3.

| | DenseNet-BC-100 | | |
|---|---|---|---|
| **Out-of-distribution datasets** | **CIFAR-10** | **CIFAR-80** | **CIFAR-100** |
| **TinyImageNet (crop)** | 1000 | 1000 | 1000 |
| **TinyImageNet (resize)** | 1000 | 1000 | 1000 |
| **LSUN (crop)** | 1000 | 1000 | 1000 |
| **LSUN (resize)** | 1000 | 1000 | 1000 |
| **iSUN** | 1000 | 1000 | 1000 |
| **Uniform** | 1000 | 1 | 1 |
| **Gaussian** | 1000 | 1 | 1 |
| **CIFAR-20** | - | 1 | - |

Table 5: Optimal Temperature $T$ for reproducing main results in Table 2 and 3.

| | Wide-ResNet-28-10 | | |
|---|---|---|---|
| **Out-of-distribution datasets** | **CIFAR-10** | **CIFAR-80** | **CIFAR-100** |
| **TinyImageNet (crop)** | 0.0005 | 0.0002 | 0.0026 |
| **TinyImageNet (resize)** | 0.0011 | 0.0004 | 0.0024 |
| **LSUN (crop)** | 0 | 0.0002 | 0.0038 |
| **LSUN (resize)** | 0.0006 | 0.0002 | 0.0026 |
| **iSUN** | 0.0008 | 0.0002 | 0.0026 |
| **Uniform** | 0.0014 | 0.0002 | 0.0032 |
| **Gaussian** | 0.0014 | 0.0002 | 0.0032 |
| **CIFAR-20** | - | 5e-05 | - |

Table 6: Optimal perturbation magnitude $\varepsilon$ for reproducing main results in Table 2 and 3.

| | Wide-ResNet-28-10 | | |
| Out-of-distribution datasets | CIFAR-10 | CIFAR-80 | CIFAR-100 |
|---|---|---|---|
| **TinyImageNet (crop)** | 1000 | 1000 | 1000 |
| **TinyImageNet (resize)** | 1000 | 1000 | 1000 |
| **LSUN (crop)** | 1000 | 1000 | 1000 |
| **LSUN (resize)** | 1000 | 1000 | 1000 |
| **iSUN** | 1000 | 1000 | 1000 |
| **Uniform** | 1000 | 1 | 1000 |
| **Gaussian** | 1000 | 1 | 1000 |
| **CIFAR-20** | - | 1 | - |

Table 7: Optimal Temperature $T$ for reproducing main results in Table 2 and 3.

| | Wide-ResNet-40-4 | | |
| Out-of-distribution datasets | CIFAR-10 | CIFAR-80 | CIFAR-100 |
|---|---|---|---|
| **TinyImageNet (crop)** | 0.0004 | 0.0002 | 0.0014 |
| **TinyImageNet (resize)** | 0.0008 | 0.0004 | 0.0016 |
| **LSUN (crop)** | 0 | 0.0002 | 0.0038 |
| **LSUN (resize)** | 0.001 | 0.0002 | 0.0014 |
| **iSUN** | 0.0008 | 0.0002 | 0.0016 |
| **Uniform** | 0.0016 | 0.0002 | 0.0024 |
| **Gaussian** | 0.0016 | 0.0002 | 0.0026 |
| **CIFAR-20** | - | 0.0002 | - |

Table 8: Optimal perturbation magnitude $\varepsilon$ for reproducing main results in Table 2 and 3.

| | Wide-ResNet-40-4 | | |
| Out-of-distribution datasets | CIFAR-10 | CIFAR-80 | CIFAR-100 |
|---|---|---|---|
| **TinyImageNet (crop)** | 1000 | 1000 | 1000 |
| **TinyImageNet (resize)** | 1000 | 1000 | 1000 |
| **LSUN (crop)** | 1000 | 1000 | 1000 |
| **LSUN (resize)** | 1000 | 1000 | 1000 |
| **iSUN** | 1000 | 1000 | 1000 |
| **Uniform** | 1000 | 1 | 1000 |
| **Gaussian** | 1000 | 1 | 1000 |
| **CIFAR-20** | - | 1 | - |

Table 9: Optimal Temperature $T$ for reproducing main results in Table 2 and 3.

# B Supplementary Results in Section 5.1

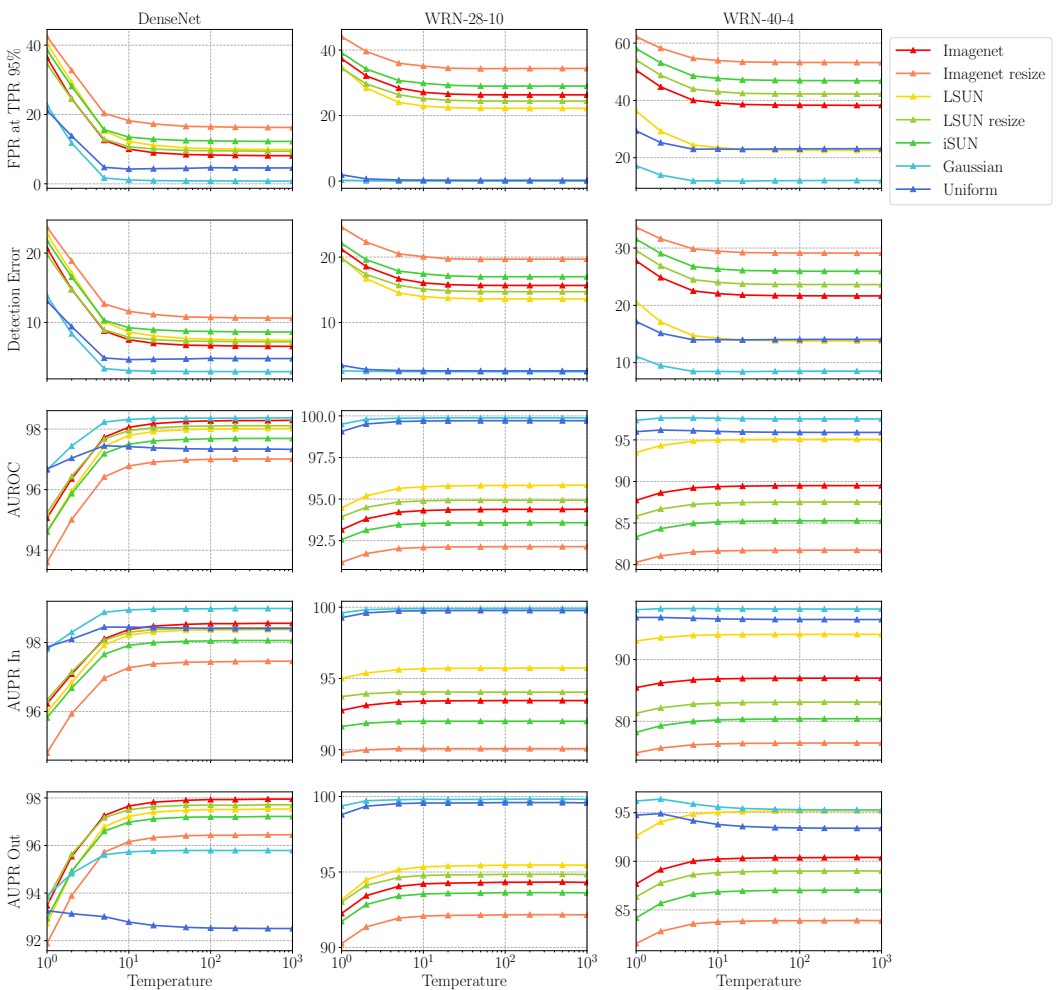

Figure 7: Detection performance on DenseNet, Wide ResNet-28-10 and Wide ResNet-40-4 under different temperature, when input preprocessing is not used, i.e., $\varepsilon = 0$. All networks are trained on CIFAR-10.

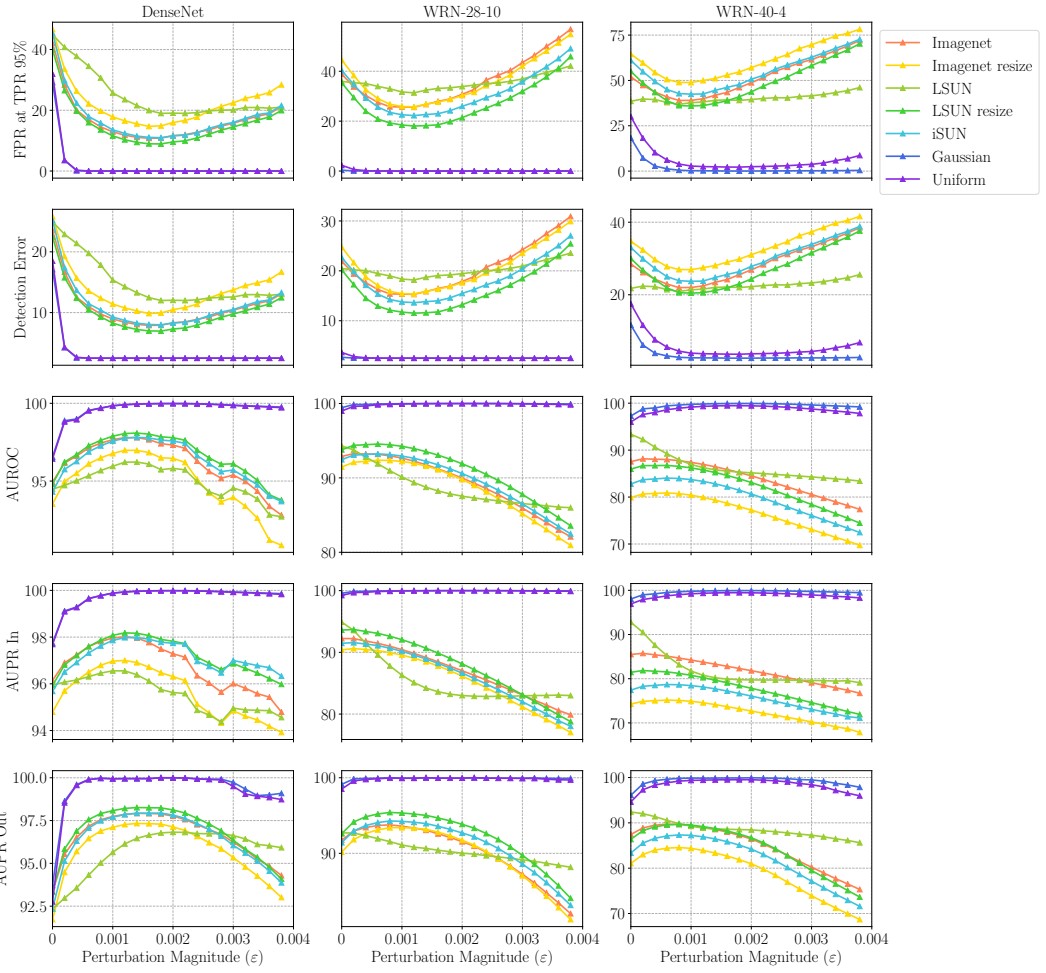

Figure 8: Detection performance on DenseNet, Wide ResNet-28-10 and Wide ResNet-40-4 under different perturbation magnitude, when temperature scaling is not used, i.e., $T = 1$. All networks are trained on CIFAR-10.

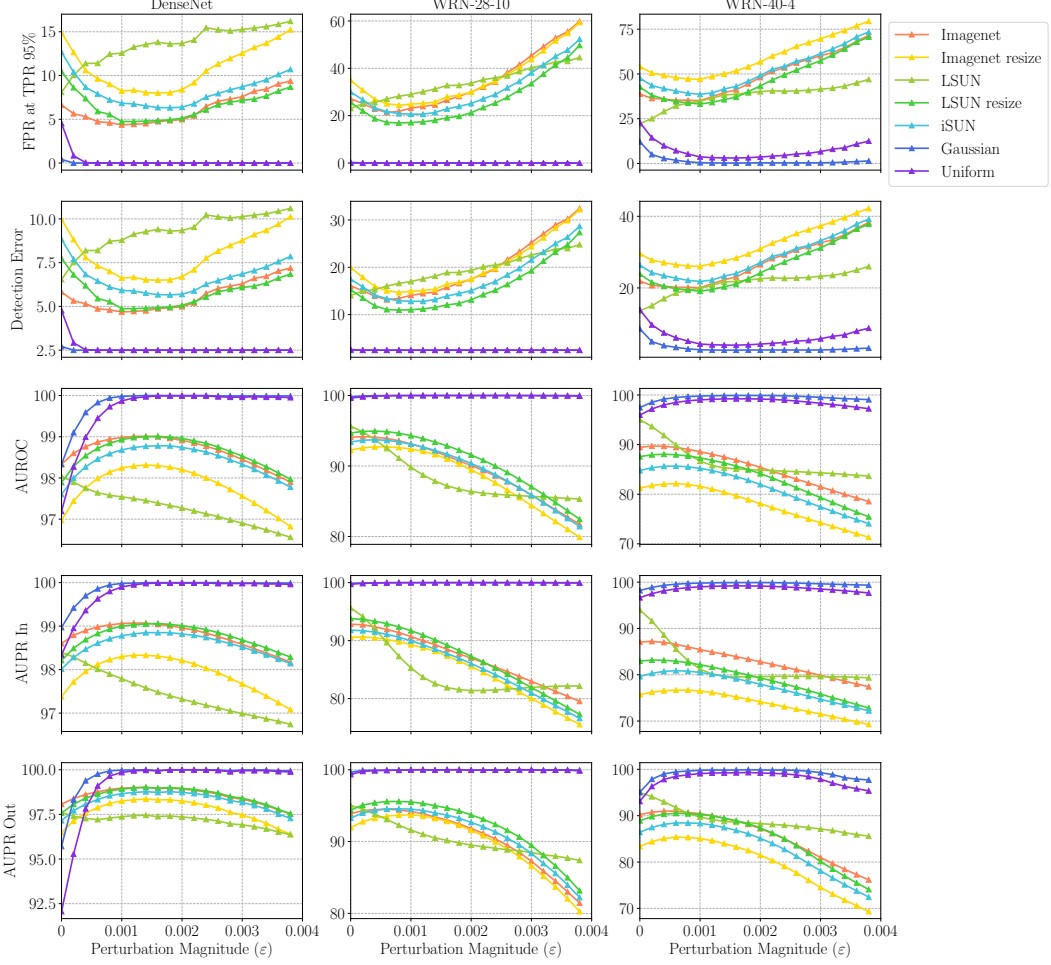

Figure 9: Detection performance on DenseNet, Wide ResNet-28-10 and Wide ResNet-40-4 under different perturbation magnitude, when the optimal temperature is used, i.e., $T = 1000$. All networks are trained on CIFAR-10.

## C SUPPLEMENTARY RESULTS IN SECTION 5.2 AND 5.3

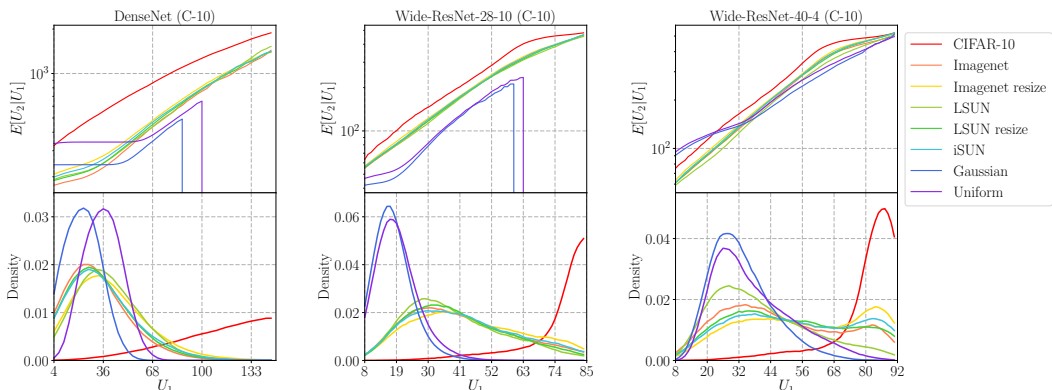

Figure 10: Expectation of the second order term $U_2$ conditioned on the first order term $U_1$ under DenseNet, Wide-ResNet-28-10 and Wide ResNet-40-4. All networks are trained on CIFAR-10.

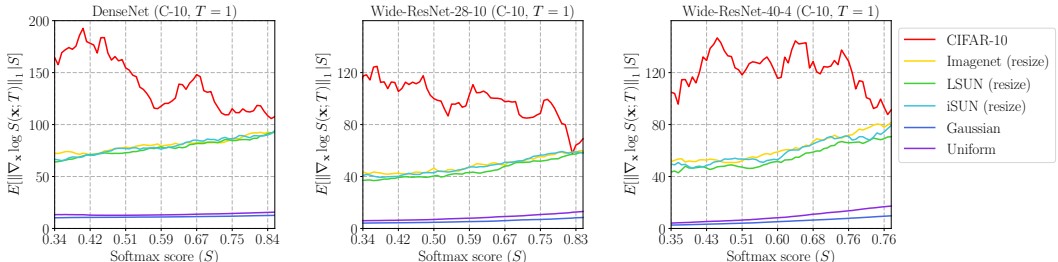

Figure 11: Expectation of gradient norms conditioned on the softmax scores under DenseNet, Wide-ResNet-28-10 and Wide ResNet-40-4, where the temperature scaling is not used. All networks are trained on CIFAR-10.

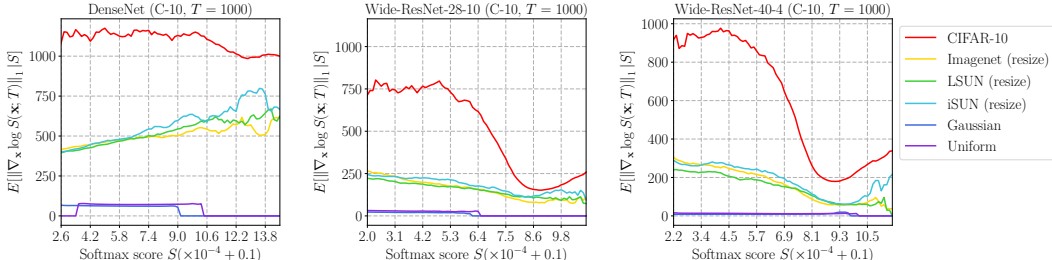

Figure 12: Expectation of gradient norms conditioned on the softmax scores under DenseNet, Wide-ResNet-28-10 and Wide ResNet-40-4, where the optimal temperature is used, i.e., $T = 1000$. All networks are trained on CIFAR-10.

# D   TAYLOR EXPANSION

In this section, we present the Taylor expansion of the soft-max score function:

$$
S_{\hat{y}}(\boldsymbol{x};T) = \frac{\exp\left(f_{\hat{y}}(\boldsymbol{x})/T\right)}{\sum_{i=1}^{N}\exp(f_i(\boldsymbol{x})/T)}
$$

$$
= \frac{1}{\sum_{i=1}^{N}\exp\left(\frac{f_i(\boldsymbol{x})-f_{\hat{y}}(\boldsymbol{x})}{T}\right)}
$$

$$
= \frac{1}{\sum_{i=1}^{N}\left[1+\frac{f_i(\boldsymbol{x})-f_{\hat{y}}(\boldsymbol{x})}{T}+\frac{1}{2!}\frac{(f_i(\boldsymbol{x})-f_{\hat{y}}(\boldsymbol{x}))^2}{T^2}+o\left(\frac{1}{T^2}\right)\right]} \qquad \text{by Taylor expansion}
$$

$$
\approx \frac{1}{N-\frac{1}{T}\sum_{i=1}^{N}[f_{\hat{y}}(\boldsymbol{x})-f_i(\boldsymbol{x})]+\frac{1}{2T^2}\sum_{i=1}^{N}[f_i(\boldsymbol{x})-f_{\hat{y}}(\boldsymbol{x})]^2}
$$

# E   PROPOSITION 1

The following proposition 1 shows that the detection error $P_e(T,0) \approx c$ if $T$ is sufficiently large. Thus, increasing the temperature further can only slightly improve the detection performance.

**Proposition 1.** *There exists a constant $c$ only depending on function $U_1$, in-distribution $P_{\boldsymbol{X}}$ and out-of-distribution $Q_{\boldsymbol{X}}$ such that $\lim_{T\to\infty} P_e(T,\varepsilon) = c$, when $\varepsilon = 0$ (i.e., no input preprocessing).*

*Proof.* Since

$$
S_{\hat{y}}(\boldsymbol{X};T) = \frac{\exp(f_{\hat{y}}(\boldsymbol{X})/T)}{\sum_{i=1}^{N}\exp(f_i(\boldsymbol{X})/T)} = \frac{1}{1+\sum_{i\neq\hat{y}}\exp([f_i(\boldsymbol{X})-f_{\hat{y}}(\boldsymbol{X})]/T)}
$$

Therefore, for any $\boldsymbol{X}$,

$$
\lim_{T\to\infty} T\left(-\frac{1}{S_{\hat{y}}(\boldsymbol{X};T)}+N\right) = \lim_{T\to\infty}\sum_{i\neq\hat{y}} T\left[1-\exp\left(\frac{f_i(\boldsymbol{X})-f_{\hat{y}}(\boldsymbol{X})}{T}\right)\right]
$$

$$
= \sum_{i\neq\hat{y}}[f_{\hat{y}}(\boldsymbol{X})-f_i(\boldsymbol{X})] = (N-1)U_1(\boldsymbol{X})
$$

This indicates that the random variable

$$
T\left(-\frac{1}{S_{\hat{y}}(\boldsymbol{X};T)}+N\right) \to (N-1)U_1(\boldsymbol{X}) \quad a.s.
$$

as $T \to \infty$. This means that for a specific $\alpha > 0$, choosing the threshold $\delta_T = 1/(N-\alpha/T)$, then the false positive rate

$$
\mathrm{FPR}(T) = Q_{\boldsymbol{X}}(S_{\hat{y}}(\boldsymbol{X};T) > 1/(N-\alpha/T)) = Q_{\boldsymbol{X}}\left(T\left(N-\frac{1}{S_{\hat{y}}(\boldsymbol{X};T)}\right) > \alpha\right)
$$

$$
\xrightarrow{T\to\infty} Q_{\boldsymbol{X}}\left((N-1)U_1(\boldsymbol{X}) > \alpha\right),
$$

and the true positive rate

$$
\mathrm{TPR}(T) = P_{\boldsymbol{X}}(S_{\hat{y}}(\boldsymbol{X};T) > 1/(N-\alpha/T)) = P_{\boldsymbol{X}}\left(T\left(N-\frac{1}{S_{\hat{y}}(\boldsymbol{X};T)}\right) > \alpha\right)
$$

$$
\xrightarrow{T\to\infty} P_{\boldsymbol{X}}\left((N-1)U_1(\boldsymbol{X}) > \alpha\right).
$$

Choosing $\alpha^*$ such that $P_{\boldsymbol{X}}\left((N-1)U_1(\boldsymbol{X}) > \alpha^*\right) = 0.95$, then $\mathrm{TPR}(T) \to 0.95$ as $T \to \infty$ and at the same time $\mathrm{FPR}(T) \to Q_{\boldsymbol{X}}\left((N-1)U_1(\boldsymbol{X}) > \alpha^*\right)$ as $T \to \infty$. There exists a constant $c$ depending on $U_1, P_{\boldsymbol{X}}, Q_{\boldsymbol{X}}$ and $P_Z$, such that

$$
\lim_{T\to\infty} P_e(T,0) = 0.05 P(Z=0) + P(Z=1)Q_{\boldsymbol{X}}\left((N-1)U_1(\boldsymbol{X}) > \alpha^*\right) = c.
$$

$\square$

# F ANALYSIS OF TEMPERATURE

For simplicity of the notations, let $\Delta_i = f_{\hat{y}} - f_i$ and thus $\Delta = \{\Delta_i\}_{i \neq \hat{y}}$. Besides, let $\bar{\Delta}$ denote the mean of the set $\Delta$. Therefore,

$$\bar{\Delta} = \frac{1}{N-1} \sum_{i \neq \hat{y}} \Delta_i = \frac{1}{N-1} \sum_{i \neq \hat{y}} [f_{\hat{y}} - f_i] = U_1.$$

Equivalently,

$$U_1 = \text{Mean}(\Delta).$$

Next, we will show

$$U_2 = \frac{1}{N-1} \sum_{i \neq \hat{y}} [f_{\hat{y}} - f_i]^2 = \overbrace{\frac{1}{N-1} \sum_{i \neq \hat{y}} [\Delta_i - \bar{\Delta}]^2}^{\text{Variance}^2(\Delta)} + \overbrace{\bar{\Delta}^2}^{\text{Mean}^2(\Delta)}.$$

Since

$$
\begin{aligned}
U_2 &= \frac{1}{N-1} \sum_{i \neq \hat{y}} \Delta_i^2 && \text{by} \Delta_i = f_{\hat{y}} - f_i \\
&= \frac{1}{N-1} \sum_{i \neq \hat{y}} (\Delta_i - \bar{\Delta} + \bar{\Delta})^2 \\
&= \frac{1}{N-1} \sum_{i \neq \hat{y}} [(\Delta_i - \bar{\Delta})^2 - 2(\Delta_i - \bar{\Delta})\bar{\Delta} + \bar{\Delta}^2] \\
&= \underbrace{\frac{1}{N-1} \sum_{i \neq \hat{y}} [\Delta_i - \bar{\Delta}]^2}_{\text{Variance}^2(\Delta)} - \underbrace{\frac{2\bar{\Delta}}{N-1} \sum_{i \neq \hat{y}} (\Delta_i - \bar{\Delta})}_{=0} + \underbrace{\bar{\Delta}^2}_{\text{Mean}^2(\Delta)}
\end{aligned}
$$

then

$$U_2 = \text{Variance}^2(\Delta) + \text{Mean}^2(\Delta)$$

# G ADDITIONAL RESULTS IN SECTION 4.5

Apart from the Maximum Mean Discrepancy, we as well calculate the Energy distance between in- and out-of-distribution datasets. Let $P$ and $Q$ denote two different distributions. Then the energy distance between distributions $P$ and $Q$ is defined as

$$D^2_{\text{energy}}(P, Q) = 2\mathbb{E}_{V \sim P, W \sim Q} \|X - Y\| - \mathbb{E}_{V, V' \sim P} \|X - X'\| - \mathbb{E}_{W, W' \sim Q} \|Y - Y'\|.$$

Therefore, the energy distance between two datasets $V = \{V_1, ..., V_m\} \overset{iid}{\sim} P$ and $W = \{W_1, ..., W_m\} \overset{iid}{\sim} Q$ is defined as

$$\widehat{D_{\text{energy}}}^2(P, Q) = \frac{2}{m^2} \sum_{i=1}^{m} \sum_{j=1}^{m} \|V_i - W_j\| - \frac{1}{\binom{m}{2}} \sum_{i \neq j} \|V_i - V_j\| - \frac{1}{\binom{m}{2}} \sum_{i \neq j} \|W_i - W_j\|.$$

In the experiment, we use the 2-norm $\| \cdot \|_2$.

| In-distribution datasets | Out-of-distribution Datasets | MMD Distance | Energy Distance |
|---|---|---|---|
| CIFAR-100 | Tiny-ImageNet (crop) | 0.41 | 2.25 |
| | LSUN (crop) | 0.43 | 2.31 |
| | Tiny-ImageNet (resize) | 0.088 | 0.54 |
| | LSUN (resize) | 0.12 | 0.63 |
| | iSUN (resize) | 0.11 | 0.56 |
| CIFAR-80 | Tiny-ImageNet (crop) | 0.4 | 2.22 |
| | LSUN (crop) | 0.43 | 2.29 |
| | Tiny-ImageNet (resize) | 0.095 | 0.57 |
| | LSUN (resize) | 0.120 | 0.62 |
| | iSUN (resize) | 0.116 | 0.61 |
| | CIFAR-20 | 0.057 | 0.35 |

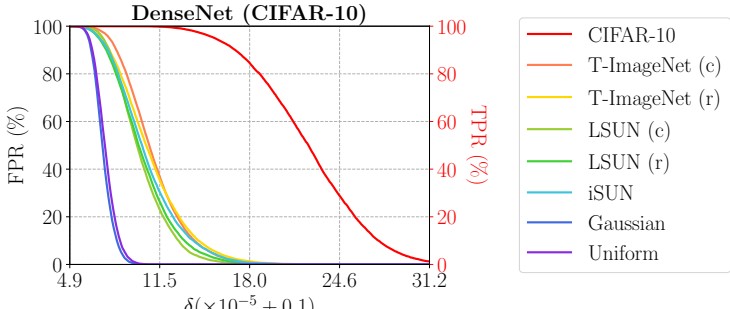

Figure 13: False positive rate (FPR) and true positive rate (TPR) under different thresholds ($\delta$) when the temperature ($T$) is set to $1,000$ and the perturbation magnitude ($\varepsilon$) is set to $0.0014$. The DenseNet is trained on CIFAR-10.

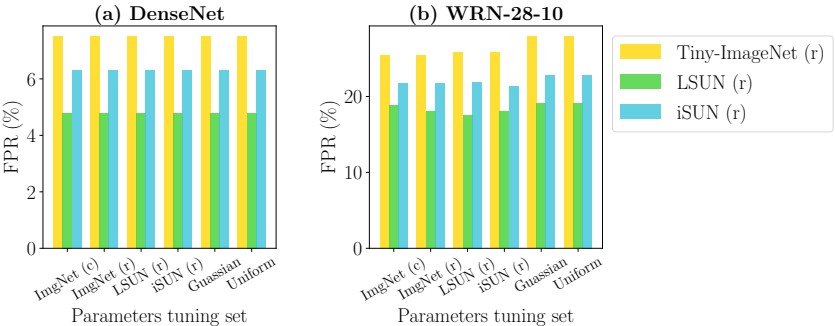

Figure 14: Detection performance on Tiny-ImageNet (resize), LSUN (resize) and iSUN (resize) when parameters are tuned on six different out-of-distribution datasets. Each tuning set contains 1,000 images and each test set contains 9,000 images. Both DenseNet and Wide-ResNet are trained on CIFAR-10. Additional results on other datasets are provide in Table 10 and 11.

## H  ADDITIONAL DISCUSSIONS

In this section, we present additional discussion on the proposed method. We first empirically show how the threshold $\delta$ affects the detection performance. We next show how the proposed method performs when the parameters are tuned on a certain out-of-distribution dataset and are evaluated on other out-of-distribution datasets. Finally, we show how the size of dataset for choosing parameters affects the detection performance.

**Effects of the threshold.** We analyze how the threshold affects the following metrics: (1) FPR, i.e., the fraction of out-of-distribution images misclassified as in-distribution images; (2) TPR, i.e, the fraction of in-distribution images correctly classified as in-distribution images. In Figure 13, we show how the thresholds affect FPR and TPR when the temperature and perturbation magnitude are chosen optimally (i.e., $T = 1,000$, $\varepsilon = 0.0014$). From the figure, we can observe that the threshold corresponding to 95% TPR can produce small FPRs on all out-of-distribution datasets.

**Performance across datasets.** To investigate how the parameters generalize across datasets, we tune the parameters using one out-of-distribution dataset and then evaluate on a different one. Given an out-of-distribution dataset, we first split the dataset into two disjoint subsets: **tuning set** and **test set**. The tuning set contains 1,000 images and the test set contains 9,000 images. We tune the parameters on the tuning set and evaluate the detection performance on the test set. We first choose the temperature $T$ and the perturbation magnitude $\varepsilon$ such that the FPR at TPR 95% is minimized on the tuning set of one out-of-distribution dataset. Next, we set $\delta$ to the threshold corresponding to 95% TPR and calculate the false positive rates on the test sets of other out-of-distribution datasets.

In Figure 14, we show the detection performance on three out-of-distribution datasets when the parameters are tuned on six different datasets. From Figure 14, we can observe that the parameters tuned on different tuning sets can have quite similar detection FPRs on all of three out-of-distribution image sets. This may be due to the fact, shown in Figure 13, that the threshold corresponding to 95% TPR can produce small FPRs on all datasets.

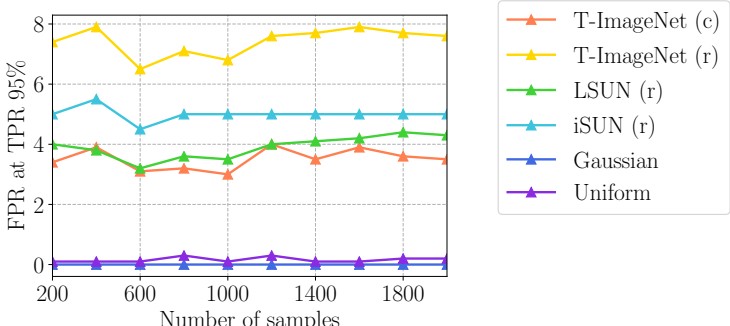

Figure 15: FPR at TPR 95% under different tuning set sizes. The DenseNet is trained on CIFAR-10 and each test set contains 8,000 out-of-distribution images.

**Performance vs. tuning set size.** To show the effects of the tuning set size on the detection performance, we devise the following experiment. For each out-of-distribution dataset, we choose the tuning set size from $200, 400, 600, 800, 1000, 1200, 1400, 1600, 1800, 2000$. For each set size, we tune the temperature $T$ and perturbation magnitude $\varepsilon$ to minimize the FPR at TPR 95% and calculate the FPR. In Figure 15, we show the detection performance of ODIN under different tuning set size. From Figure 15, we can observe that the FPR at TPR 95% tends to stabilize when the set size grows above 1,000.

| | DenseNet-BC-100 | | | | | | |
|---|---|---|---|---|---|---|---|
| Test set | ImgNet (c) | ImgNet (r) | LSUN (c) | LSUN (c) | iSUN | Gaussian | Uniform |
| | Baseline (Hendrycks & Gimpel, 2017) / Ours | | | | | | |
| ImgNet (c) | 34.7/4.3 | 34.7/4.3 | 34.7/6.6 | 34.7/4.3 | 34.7/4.3 | 34.7/4.3 | 34.7/4.3 |
| ImgNet (r) | 40.7/7.5 | 40.7/7.5 | 40.7/14.9 | 40.7/7.5 | 40.7/7.5 | 40.7/7.5 | 40.7/7.5 |
| LSUN (c) | 39.3/13.8 | 39.3/13.8 | 39.3/8.1 | 39.3/13.8 | 39.3/13.8 | 39.3/13.8 | 39.3/13.8 |
| LSUN (r) | 33.6/4.8 | 33.6/4.8 | 33.6/10.4 | 33.6/4.8 | 33.6/4.8 | 33.6/4.8 | 33.6/4.8 |
| iSUN | 37.2/6.3 | 37.2/6.3 | 37.2/12.6 | 37.2/6.3 | 37.2/6.3 | 37.2/6.3 | 37.2/6.3 |
| Gaussian | 23.5/0.0 | 23.5/0.0 | 23.5/0.4 | 23.5/0.0 | 23.5/0.0 | 23.5/0.0 | 23.5/0.0 |
| Uniform | 12.3/0.0 | 12.3/0.0 | 12.3/4.5 | 12.3/0.0 | 12.3/0.0 | 12.3/0.0 | 12.3/0.0 |

Table 10: Detection performance across different datasets. Each row corresponds to the FPR at TPR 95% on the same test set where parameters are tuned under different tuning sets. Each column corresponds to the FPR at TPR 95% on different test sets where parameters are tuned under the same tuning set. The DenseNet is trained on CIFAR-10.

| | Wide-ResNet- 28-10 | | | | | | |
|---|---|---|---|---|---|---|---|
| Test set | ImgNet (c) | ImgNet (r) | LSUN (c) | LSUN (c) | iSUN | Gaussian | Uniform |
| | Baseline (Hendrycks & Gimpel, 2017) / Ours | | | | | | |
| ImgNet (c) | 38.9/23.4 | 38.9/24.5 | 38.9/27.1 | 38.9/23.4 | 38.9/24.1 | 38.9/26.5 | 38.9/26.5 |
| ImgNet (r) | 45.6/25.5 | 45.6/25.5 | 45.6/32.9 | 45.6/25.8 | 45.6/25.8 | 45.6/27.9 | 45.6/27.9 |
| LSUN (c) | 35.0/28.1 | 35.0/31.27 | 35.0/21.8 | 35.0/28.2 | 35.0/28.9 | 35.0/29.7 | 35.0/29.7 |
| LSUN (r) | 35.0/18.9 | 35.0/18.0 | 35.0/25.6 | 35.0/17.6 | 35.0/18.0 | 35.0/19.1 | 35.0/19.1 |
| iSUN | 40.6/21.8 | 40.6/21.7 | 40.6/28.9 | 40.6/ 21.9 | 40.6/21.3 | 40.6/22.8 | 40.6/22.8 |
| Gaussian | 1.6/0.0 | 1.6/0.0 | 1.6/0.4 | 1.6/0.0 | 1.6/0.0 | 1.6/0.0 | 1.6/0.0 |
| Uniform | 0.3/0.0 | 0.3/0.0 | 0.3/0.0 | 0.3/0.0 | 0.3/0.0 | 0.3/0.0 | 0.3/0.0 |

Table 11: Detection performance across different datasets. Each row corresponds to the FPR at TPR 95% on the same test set where parameters are tuned under different tuning sets. Each column corresponds to the FPR at TPR 95% on different test sets where parameters are tuned under the same tuning set. The Wide-ResNet is trained on CIFAR-10.

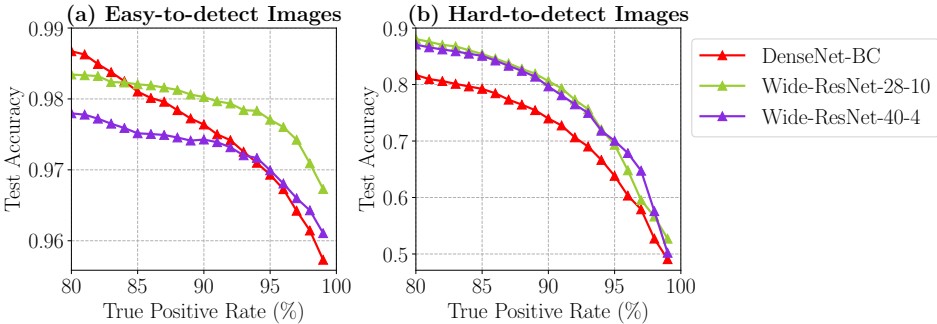

Figure 16: (a) The test accuracy on the images having softmax scores above the threshold corresponding to a certain true positive rate. (b) The test accuracy on the images having softmax scores below the threshold corresponding to a certain true positive rate. All networks are trained on CIFAR-10.

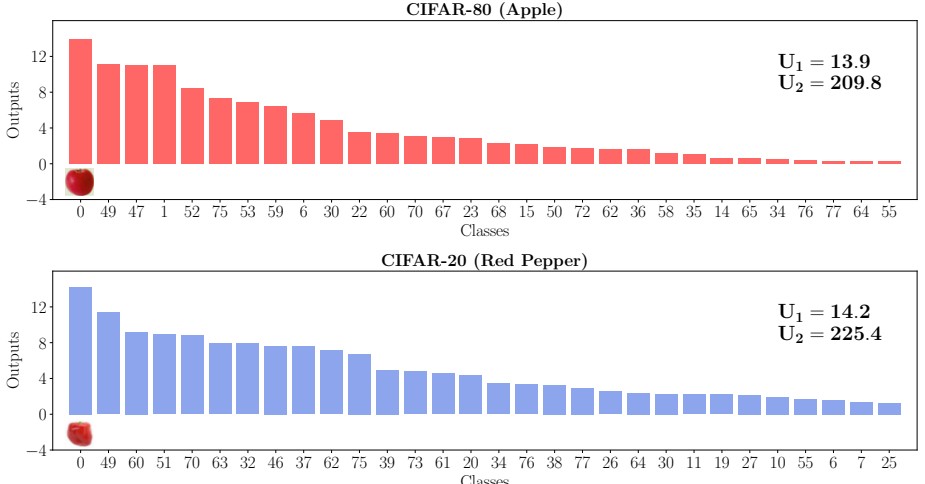

Figure 17: Outputs of DenseNet on thirty classes for an image of apple from CIFAR-80 and an image of red pepper from CIFAR-20. The label "0" denotes the class "apple" and the label "49" denotes the class "orange".

# I    ADDITIONAL ANALYSIS

**Difficult-to-classify images and difficult-to-detect images.** We analyze the correlation between the images that tend to be out-of-distribution and images on which the neural network tend to make incorrect predictions. To understand the correlation, we devise the following experiment. For the fixed temperature $T$ and perturbation magnitude $\varepsilon$, we first set $\delta$ to the softmax score threshold corresponding to a certain true positive rate. Next, we calculate the test accuracy on the images with softmax scores above $\delta$ and the test accuracy on the images with softmax score below $\delta$, respectively. We report the results in Figure 16(a) and (b). From these two figures, we can observe that the images that are difficult to detect are more likely to be the images that are difficult to classify. For example, the DenseNet can achieve up to 98.5% test accuracy on the images having softmax scores above the threshold corresponding to 80% TPR, but can only achieve around 82% test accuracy on the images having softmax scores below the threshold corresponding to 80% TPR.

**Same manifold datasets.** We provide additional empirical results showing how the term $E[U_2|U_1]$ affects the detection performance when in- and out-of-distribution datasets locate on the same manifold. In Figure 17, we show the outputs of DenseNet on thirty classes for an image of apple from CIFAR-80 (in-distribution) and an image of red pepper of CIFAR-20 (out-distribution). We can observe that the outputs of DenseNet for both images are quite similar to each other. In addition, we can observe that for both images, the second and third largest output are quite close to the largest output. This may be due the fact the image of red pepper shares some common features with the images in CIFAR-80. Furthermore, the similarity between the outputs for the images from

| | Out-of-distribution dataset | FPR (95% TPR) ↓ | Detection Error ↓ | AUROC ↑ | AUPR In ↑ | AUPR Out ↑ |
|---|---|---|---|---|---|---|
| | | Baseline (Hendrycks & Gimpel, 2017) / Ours | | | | |
| **CIFAR-10** | CIFAR-100 | 57.1/47.2 | 31.1/26.1 | 89.0/89.8 | 91.2/91.4 | 86.8/88.7 |
| **CIFAR-100** | CIFAR-10 | 81.8/81.4 | 43.4/43.2 | 76.1/76.7 | 79.9/80.4 | 71.3/72.6 |

Table 12: Distinguishing in- and out-of-distribution test set data for image classification. All values are percentages. ↑ indicates larger value is better, and ↓ indicates lower value is better. The architecture is DenseNet.

CIFAR-80 and CIFAR-20 can help explain that the detection task becomes harder when in- and out-of-distribution datasets locate on the same manifold.

**Reciprocal results between datasets.** In Table 12, we show the reciprocal results between datasets. First, we train the DenseNet on the CIFAR-10 dataset (in-distribution) and evaluate the detection performance on the CIFAR-100 dataset (out-distribution). Next, we train the DenseNet on the CIFAR-100 dataset (in-distribution) and evaluate the detection performance on the CIFAR-10 dataset (out-distribution). From Table 12, we can observe that the performance of the DenseNet trained on CIFAR-10 is better than the performance of the DenseNet trained on CIFAR-100. This may be due to the fact that the DenseNet has a higher test accuracy on CIFAR-10 (around 95%) compared to the test accuracy on CIFAR-100 (around 77%).

