# OpenReview forum: "Enhancing The Reliability of Out-of-distribution Image Detection in Neural Networks"
_ICLR.cc/2018/Conference — Accept (Poster)_

### Official Review · AnonReviewer1 · 2017-11-24

**Rating:** 6
**Confidence:** 4

**Review:**


-----UPDATE------

The authors addressed my concerns satisfactorily. Given this and the other reviews I have bumped up my score from a 5 to a 6.

----------------------


This paper introduces two modifications that allow neural networks to be better at distinguishing between in- and out- of distribution examples: (i) adding a high temperature to the softmax, and (ii) adding adversarial perturbations to the inputs. This is a novel use of existing methods.

Some roughly chronological comments follow:

In the abstract you don't mention that the result given is when CIFAR-10 is mixed with TinyImageNet.

The paper is quite well written aside from some grammatical issues. In particular, articles are frequently missing from nouns. Some sentences need rewriting (e.g. in 4.1 "which is as well used by Hendrycks...", in 5.2 "performance becomes unchanged").

 It is perhaps slightly unnecessary to give a name to your approach (ODIN) but in a world where there are hundreds of different kinds of GANs you could be forgiven.

I'm not convinced that the performance of the network for in-distribution images is unchanged, as this would require you to be able to isolate 100% of the in-distribution images. I'm curious as to what would happen to the overall accuracy if you ignored the results for in-distribution images that appear to be out-of-distribution (e.g. by simply counting them as incorrect classifications). Would there be a correlation between difficult-to-classify images, and those that don't appear to be in distribution?

When you describe the method it relies on a threshold delta which does not appear to be explicitly mentioned again.

In terms of experimentation it would be interesting to see the reciprocal of the results between two datasets. For instance, how would a network trained on TinyImageNet cope with out-of-distribution images from CIFAR 10?

Section 4.5 felt out of place, as to me, the discussion section flowed more naturally from the experimental results. This may just be a matter of taste.

I did like the observations in 5.1 about class deviation, although then, what would happen if the out-of-distribution dataset had a similar class distribution to the in-distribution one? (This is in part, addressed in the CIFAR80 20 experiments in the appendices).

This appears to be a borderline paper, as I am concerned that the method isn't sufficiently novel (although it is a novel use of existing methods).

Pros:
- Baseline performance is exceeded by a large margin
- Novel use of adversarial perturbation and temperature
- Interesting analysis

Cons:
- Doesn't introduce and novel methods of its own
- Could do with additional experiments (as mentioned above)
- Minor grammatical errors

---

> ### Author Response · Authors · 2017-12-23
> **Response to Reviewer1**
>
> We thank the reviewer for the useful feedback. We address each point raised in detail below.
>
> R1: In the abstract you don't mention that the result given is when CIFAR-10 is mixed with TinyImageNet.
>
> We have revised the last sentence of the abstract: “For example,ODIN reduces the false positive rate from the baseline 34.7% to 4.3% on the DenseNet (applied to CIFAR-10 and Tiny-ImageNet) when the true positive rate is 95%.”
>
> R1: grammatical issues and some sentences need rewriting.
> We have rewritten those sentences, and others, in the revised version.
>
> R1: I'm not convinced that the performance of the network for in-distribution images is unchanged.
>
> Yes, the overall accuracy would have been changed if we ignored the results for in-distribution images that appear to be out-of-distribution.However, we meant to say that our method does not change the label predictions for in-distribution images, since one can always use the original image and pass it through the original neural network. We have replaced the word “performance” with “predictions” to avoid confusion.
>
> R1: Would there be a correlation between difficult-to-classify images, and those that don't appear to be in distribution?
>
> We provide empirical results on the correlation between difficult-to-classify images and difficult-to-detect images (Figure 16). We can observe that the images that are difficult to detect tend to be the images that are difficult to classify  (e.g., DenseNet can only achieve  around 50% test accuracy on the images having softmax scores below the threshold corresponding to 99% TPR,  while being able to achieve around 95.2% accuracy on the overall image set).
>
> R1: When you describe the method it relies on a threshold delta which does not appear to be explicitly mentioned again.
>
> We have extensively studied the effect of $\delta$ and have provided additional results in Appendix H. Also, as mentioned in the response to Reviewer 2, we no longer optimize over delta.
>
> R1: The reciprocal of the results between two datasets.
>
> We provide the reciprocal of the results between CIFAR-10 and CIFAR-100 in Appendix I.  DenseNet can achieve 47.2% FPR at TPR 95% when the CIFAR-10 dataset is the in-distribution dataset and CIFAR-100 dataset is the out-of-distribution dataset, while achieving 81.4% FPR at TPR 95% when the CIFAR-100 dataset is the in-distribution and CIFAR-10 dataset is the out-of-distribution dataset.
>
> R1: What would happen if the out-of-distribution dataset had a similar class distribution to the in-distribution one?
>
> We provide additional results in Appendix I (Figure 17), where we show the outputs of DenseNet on thirty classes for an image of apple from CIFAR-80 (in-distribution) and an image of red pepper from CIFAR-20 (out-distribution). We can observe that when the out-of-distribution images share a few common features with the in-distribution images (e.g., the image of apple is quite similar to the image of red pepper), the output distribution of the neural networks for the out-of-distribution images are sometimes similar to the output distribution for the in-distribution images.
>
> R1: The method isn't sufficiently novel (although it is a novel use of existing methods).
>
> Our proposed method is inspired by the existing methods used in other tasks (temperature scaling used for distilling the knowledge in neural networks, Hinton et al., 2015, and adding small perturbations used for generating adversarial examples, Goodfellow et al. 2015). What is novel is the way in which we use perturbation: we do exactly the opposite of what Goodfellow et al. 2015 do; instead of adding, we actually subtract the perturbation suggested them. The fact that this, along with the temperature scaling, improves the out-of-distribution detection performance is surprising and novel. Further, our work also has merits in providing extensive experimental analysis and theoretical insights, and justifying the novel use case of these techniques for out-of-distribution image detection.

---

### Official Review · AnonReviewer2 · 2017-11-27
**simple, effective technique**

**Rating:** 6
**Confidence:** 3

**Review:**

The paper proposes a new method for detecting out of distribution samples. The core idea is two fold: when passing a new image through the (already trained) classifier, first preprocess the image by adding a small perturbation to the image pushing it closer to the highest softmax output and second, add a temperature to the softmax. Then, a simple decision is made based on the output of the softmax of the perturbed image - if it is able some threshold then the image is considered in-distribution otherwise out-distribution.

This paper is well written, easy to understand and presents a simple and apparently effective method of detecting out of distribution samples. The authors evaluate on cifar-10/100 and several out of distribution datasets and this method outperforms the baseline by significant margins. They also examine the effects of the temperature and step size of the perturbation.

My only concern is that the parameter delta (threshold used to determine in/out distribution) is not discussed much. They seem to optimize over this parameter, but this requires access to the out of distribution set prior to the final evaluation. Could the authors comment on how sensitive the method is to this parameter? How much of the out of distribution dataset is used to determine this value, and what are the effects of this size during tuning? What happens if you set the threshold using one out of distribution dataset and then evaluate on a different one? This seems to be the central part missing to this paper and if the authors are able to address it satisfactorily I will increase my score.

---

> ### Author Response · Authors · 2017-12-23
> **Response to Reviewer2**
>
> We thank Reviewer 2 for the constructive and encouraging feedback!
>
> To address your concern about delta, we are no longer optimizing with respect to this parameter. We tune the temperature (T), perturbation magnitude ($\epsilon$) on an out-of-distribution image set (for a given in-distribution image set) and setting $\delta$ to the threshold corresponding to the 95% TPR.  Our experiments in Appendix H appears to indicate that the choice of the out-of-distribution image set used to tune the parameters does not matter very much. Our method shows superior performance compared to the state-of-the-art whether we use Tiny-ImageNet (cropped) or Tiny-ImageNet (resized) or LSUN (resized) or iSUN (resized) or Gaussian noise or Uniform noise as the out-of-distribution  dataset during the parameter tuning process. We also note that, while we may use one of these datasets during the tuning process, the testing is performed against other out-of-distribution dataset as well.
>
> Following the suggestions, we extensively studied the effect of $\delta$ and thereafter summarize our findings below.
> How sensitive the method is to the threshold?
> In Figure 13, we show how the thresholds affect FPR and TPR, where we can observe that the threshold corresponding to 95% TPR can produce small FPRs on all out-of-distribution datasets.
>
> (2) How much of the out of distribution dataset is used to determine this value, and what are the effects of this size during tuning?
> In the main results reported in Table 2, we held out 1,000 images to tune the parameters and evaluated on the remaining 9,000 images. To further understand the effect of the tuning set size, we show in Figure 15 the detection performance as we vary the tuning set size, ranging from 200 to 2000. We evaluate the detection performance on the remaining 8,000 images. In general we found the performance tends to stabilize as the tuning set size varies..
>
> (3) How does delta generalize across datasets?
> In addition to the observation in Figure 13 (a) and (b) that the effect of $delta$ is quite similar across datasets, we further conducted experiments as suggested by the reviewer. Specifically, we set the threshold using one out of distribution dataset and then evaluate on a different one. All the results can be found in Appendix H (Figure 14). We observe that the parameters tuned on different out-of-distribution natural image sets have quite similar detection performance.

---

### Official Review · AnonReviewer3 · 2017-11-29
**Improvement on state-of-the-art for detecting out of distribution examples**

**Rating:** 9
**Confidence:** 3

**Review:**

Detecting out of distribution examples is important since it lets you know when neural network predictions might be garbage. The paper addresses this problem with a method inspired by adversarial training, and shows significant improvement over best known method, previously published in ICLR 2017.

Previous method used at the distribution of softmax scores as the measure. Highly peaked -> confidence, spread out -> out of distribution. The authors notice that in-distribution examples are also examples where it's easy to drive the confidence up with a small step. The small step is in the direction of gradient when top class activation is taken as the objective. This is also the gradient used to determine influence of predictors, and it's the gradient term used for adversarial training "fast gradient sign" method.

Their experiments show improvement across the board using DenseNet on collection of small size dataset (tiny imagenet, cifar, lsun). For instance at 95% threshold (detect 95% of out of distribution examples), their error rate goes down from 34.7% for the best known method, to 4.3% which is significant enough to prefer their method to the previous work.

---

> ### Author Response · Authors · 2017-12-23
> **Response to Reviewer3**
>
> Thank you for the encouraging feedback on the paper.

---

### Public Comment · ~Siniša_Šegvić1 · 2018-01-04
**Clarifying interaction between \delta and T might improve the paper**

Detecting outliers by perturbing the input is intuitively clear (and very nice!). Near inliers, the model has relatively low curvature due to strong training signal. Thus, anti-adversarial perturbation of an inlier is likely to increase the probability of the dominant class. On the other hand, the model may change arbitrarily near outliers. Hence, anti-adversarial perturbations are likely to produce less dependable effects there.

However, I do not understand the advantage of simultaneously fitting the softmax temperature (T) *and* the softmax output threshold \delta. It appears as these two parameters should cancel out each other, and yet Fig.3 suggests that somehow T=1000 is better than T=1 when \delta is set for TPR=95%. Clarifying the interaction between \delta and T might improve the paper.

---

> ### Author Response · Authors · 2018-01-04
> **Thanks for your comments**
>
> Thank you for your comments.  The analysis of the effects of temperature scaling can be found in Section 5.2, where we provide some insight into why changing T and delta can improve the detection performance.
>
> Here we provide some additional explanation. Suppose we consider two images where the difference between the largest output and the average of the rest of the outputs (denoted by U_1 in Section 2) is only slightly larger for the in-distribution image than  the out-of-distribution image.  In that same section, we have denoted the variance in the output by U_2, and have used Taylor's series to suggest that the soft-max score of the largest output is then determined by U_1 divided by T and U_2 divided by T^2. We also argue and provide empirical evidence to show that U_2 is larger for in-distribution images than out of distribution images. Thus, the soft-max score of the largest output of the in-distribution image can be smaller than the soft-max score of the largest output of the out-of-distribution image. To remedy this situation, Taylor's series suggests that increasing T will reduce the impact of U_2 and U_1 will dominate. Now, as we change T, we have to correspondingly change delta (since a larger T pushes all the soft-max scores towards 1/N) to be able to detect an in-distribution and an out-of-distribution image.

---

> > ### Public Comment · ~Siniša_Šegvić1 · 2018-01-04
> > **Thanks**
> >
> > Thank you for your clarification, I had missed the discussion in 5.2 and 5.3.

---

### Public Comment · ~Ananya_Kumar1 · 2018-11-26
**T and epsilon tuned on out of distribution data**

This is a good paper. Recently (e.g. in ICLR 2019 submissions) the topic of novelty detection has been gathering more interest. A number of recent works compare with ODIN.

I think there are a couple of important revisions that should be made to this paper even though it is a year old. The paper should make it extra clear that T and epsilon are tuned on the out-of-distribution images. This is indeed mentioned in the middle of page 5, but only there - assumptions like this should be clearly stated and discussed in the introduction/setup/discussion. The baseline assumes no access to out-of-distribution images. None of the reviews seem to have noticed this assumption - the most positive review says the improved performance is "significant enough to prefer their method to the previous work." In reality, one could imagine situations where we do have access to small amounts of data from the outlier distribution, or perhaps T and epsilon can be tuned with "similar" out of distribution data, but there are many cases when we want to protect against an unknown distribution. So it is important that this is spelled out very clearly.

As I understand, the classical name for the problem in statistics is novelty detection (except in novelty detection we assume no access to the out of distribution data). It would be good to mention this in the related works section, for people tracing through the literature.

Again, this is a good paper which merited acceptance, but I think these changes would better situate this work in the field.

---

### Decision · Program_Chairs · 2018-01-29
**ICLR 2018 Conference Acceptance Decision**

**Decision:**

Accept (Poster)

**Comment:**

The reviewers agree that the method is simple, the results are quite good, and the paper is well written. The issues the reviewers brought up have been adequately addressed. There is a slight concern about novelty, however the approach will likely be quite useful in practice.